# Coherent modulation of the sea-level annual cycle in the United States by Atlantic Rossby waves

Francisco M. Calafat [ID] [1], Thomas Wahl[2], Fredrik Lindsten[3], Joanne Williams[1] & Eleanor Frajka-Williams[4]

Changes in the sea-level annual cycle (SLAC) can have profound impacts on coastal areas, including increased flooding risk and ecosystem alteration, yet little is known about the magnitude and drivers of such changes. Here we show, using novel Bayesian methods, that there are significant decadal fluctuations in the amplitude of the SLAC along the United States Gulf and Southeast coasts, including an extreme event in 2008–2009 that is likely (probability ≥68%) unprecedented in the tide-gauge record. Such fluctuations are coherent along the coast but decoupled from deep-ocean changes. Through the use of numerical and analytical ocean models, we show that the primary driver of these fluctuations involves incident Rossby waves that generate fast western-boundary waves. These Rossby waves project onto the basin-wide upper mid-ocean transport (top 1000 m) leading to a link with the SLAC, wherein larger SLAC amplitudes coincide with enhanced transport variability.

[1] National Oceanography Centre, Joseph Proudman Building, 6 Brownlow Street, Liverpool L3 5DA, UK. [2] Department of Civil, Environmental and Construction Engineering and National Center for Integrated Coastal Research, University of Central Florida, 12800 Pegasus Drive, Suite 211, Orlando 32816-2450 FL, USA. [3] Department of Information Technology, Uppsala University, Lägerhyddsv. 2, hus 2, Uppsala 752 37, Sweden. [4] Ocean and Earth Sciences, University of Southampton, European Way, Southampton SO14 3ZH, UK. Correspondence and requests for materials should be addressed to F.M.C. (email: francisco.calafat@noc.ac.uk)

The sea-level annual cycle (SLAC) can have local peak-to-peak amplitudes comparable to the global average sea-level rise over the 20th century (~16 cm). These annual variations in sea level have a profound effect on coastal areas. They affect the habitat availability, nutrient budgets, and productivity of estuaries[1,2]; enable substantial coastal erosion to an extent comparable, over a year, to that caused by a hurricane[3]; and modulate coastal groundwater dynamics and discharge[4]. In low-lying areas, large annual variations can also contribute to nuisance flooding, which occurs during clear-sky conditions due to the combination of high mean sea level and spring tides[5]. In addition, they can compound the effect of sea-level rise and expose the coastline to increased risk of flooding by raising the baseline for storm surges.

The SLAC is primarily associated with the response of the ocean-atmosphere system to changes in solar insolation by season and latitude, although it includes also a small gravitational contribution[6]. Such response is governed by a complex interplay between local and large-scale dynamics[7], and thus is highly location dependent. As a result, both the amplitude and phase of the SLAC exhibit great geographic variability[8,9]. Furthermore, since the climate system may respond nonlinearly to the periodic forcing by solar insolation, the oscillatory characteristics of the SLAC can change considerably over time. Indeed, significant temporal variations in the amplitude and phase of the SLAC have been observed in many regions around the world[10–18]. These changes in the SLAC can significantly exacerbate the effects of seasonal variations on coastal areas. Knowing how to model and predict these seasonal changes would provide crucial time to better protect coastal areas and to utilize their resources more effectively, in turn bringing great socioeconomic and environmental benefits. However, this requires a deep understanding of their underlying dynamics, which is still lacking in many regions.

The United States Gulf and Southeast coasts are particularly vulnerable to the effects of seasonal sea-level changes due to their hurricane-prone and predominantly low-lying coastal areas, yet studies focused on these regions are very limited[12,16]. Significant changes in the amplitude of the SLAC were observed in tide-gauge records from both regions, but the processes controlling these changes remain poorly understood. Multiple regression[16] and correlation[12] analyses were used to examine the relationship between the amplitude changes and several proxy variables. Low (~0.3) or non-significant correlations were found along the Atlantic coast for all the proxies considered[12]. Along the Gulf Coast, changes in the amplitude of the SLAC were found to correlate with air surface temperature for some periods but only very weakly with sea surface temperature and steric height[16], which is difficult to reconcile with sea-level theory and interpret in terms of direct causal processes. Therefore, a causal explanation of the changes in the SLAC amplitude is still lacking. Filling this gap in our knowledge is an immediate priority since it severely limits our ability to understand, model, and ultimately predict these seasonal sea-level changes.

The difficulty of finding a physical explanation arises because sea level depends on the density structure of the whole water column[7], which is set by both local and non-local dynamics. The implication is that sea-level changes are not necessarily governed by local forcing and thus the commonly used approach based on correlation/regression against surface atmospheric variables cannot establish causation and must be guided by theory and supported by basin-scale estimates of the ocean density field. This is especially true for western boundaries since they are strongly affected by remote forcing in the ocean interior[19].

Another aspect that merits consideration is the choice of the method to estimate changes in the amplitude of the SLAC. In the present context, the SLAC refers to the response of the climate system to the external periodic forcing by solar radiation. The response of a non-linear system to a periodic force is not necessarily periodic and often exhibits both amplitude and frequency modulation[20]. While approaches that assume a stationary annual cycle and analyze anomalies relative to such cycle are valid and can be successful at explaining the variability, allowing for deviations from periodicity provides an alternative view that can greatly facilitate the analysis and understanding of annual changes[21]. The most commonly used method to estimate changes in the SLAC is a harmonic least-squares fit to running windows of a selected length[10,11,15–18]. This method, however, suffers from the limitation of requiring a window of at least 5 years in order to yield robust estimates[8], which limits inference about variations at decadal or shorter timescales (a 5-year running mean attenuates the power of decadal signals by ~61%). In addition, this method does not provide estimates within half the window size from the edges of the time series, and uses information contained only within the corresponding window.

Here, we present a novel method based on Bayesian state-space modeling[22] that overcomes the issues of the windowing method, enabling estimation with unprecedented temporal resolution and robustness. We use our Bayesian method and a combination of sea-level observations, modeling, and theory to quantify changes in the amplitude of the SLAC along the Gulf and Southeast coasts of the United States, and provide a deep insight into the key drivers. We show that there are significant decadal fluctuations in the annual amplitude and identify an extreme event in 2008–2009 that is likely (probability ≥68%) unprecedented in the tide-gauge record. Such fluctuations are coherent over large distances along the coast from the Yucatan Peninsula to Cape Hatteras but they are confined to the coastal zone. The primary driver involves density anomalies propagating westward as baroclinic Rossby waves which, on reaching the western boundary, generate fast boundary waves that modulate the SLAC along the coast. These density anomalies drive changes in the geostrophic component of the meridional overturning circulation (MOC) at 26.5°N, both in observations from the Rapid Climate Change Programme[23] (RAPID) and in the ocean models, leading to a link between the SLAC and the upper mid-ocean transport (UMO, the top 1000 m meridional transport).

## Results

**Changes in the SLAC amplitude from tide-gauge records**. Time series of the SLAC amplitude for tide-gauge records along the United States Gulf and Atlantic coasts are shown in Fig. 1a (the location of the tide gauges is shown in Supplementary Fig. 1). All time series display significant amplitude variations (up to 71% of the time-mean value) at decadal timescales, reflecting strong SLAC changes. These variations show a striking regional coherence along two distinct sections of coastline divided at approximately Cape Hatteras. Amplitude changes across stations to the south of Cape Hatteras (stations 1–14) are very coherent and show both larger magnitude (up to 7.8 cm from the time mean) and a larger time-mean value (up to 11.1 cm) than changes at stations north of Cape Hatteras (stations 15–25) (time-mean value of 6.5 cm and deviations of up to 4.6 cm). This suggests that two different regimes of seasonal variability are operating north and south of Cape Hatteras. This regional coherence and the division line marked by Cape Hatteras is made clearer by plotting the correlation matrix of the amplitude time series (Fig. 1b). The cross-correlation for stations on the same side of Cape Hatteras, either south or north, is very high (average of 0.80 and 0.89, respectively) reflecting the coherence along the two coastline sections, but it is much lower (average of 0.36) for stations on

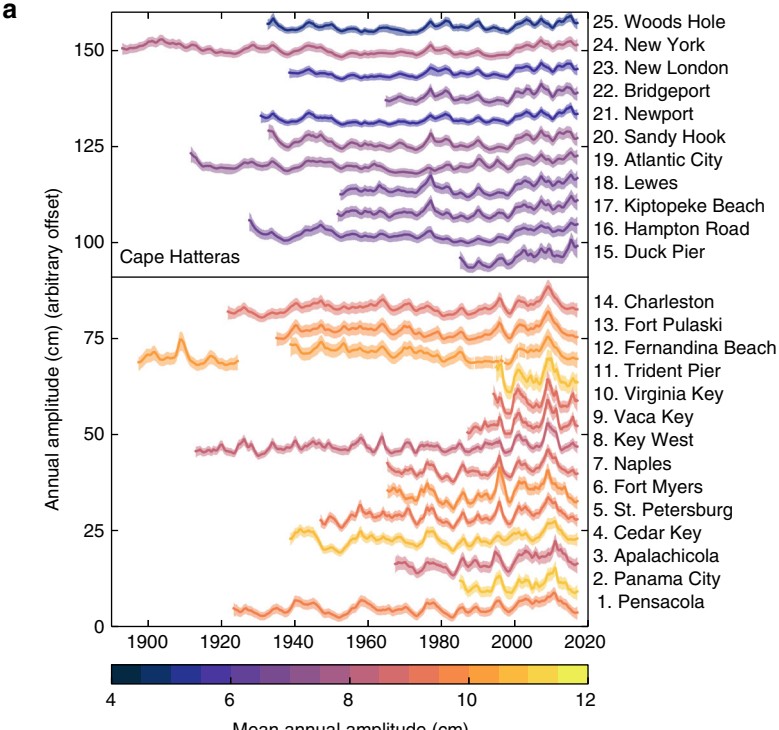

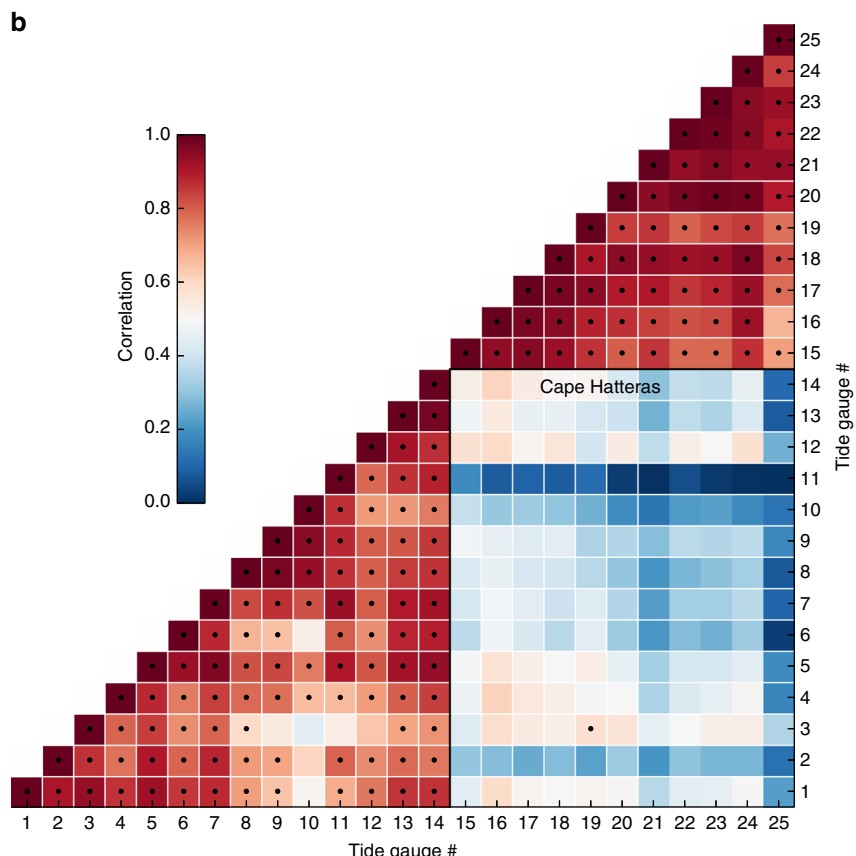

**Fig. 1** Amplitude from tide-gauge records. **a** Temporal changes in the amplitude of the SLAC from tide-gauge records along the United States Gulf and Atlantic coasts as estimated using a Bayesian state-space model. Solid lines denote the mean of the posterior distribution at each time step, whereas shaded areas represent the 68% (1-sigma) credible interval. The colors of the solid lines denote the time-mean value of the annual amplitude. The name of each station along with their identification number are also shown (see Supplementary Fig. 1 for tide-gauge locations). **b** Correlation matrix of the time series shown in **a**. Numbers along the axes represent tide-gauge identification numbers, whereas black dots denote significant correlation at the 95% confidence level

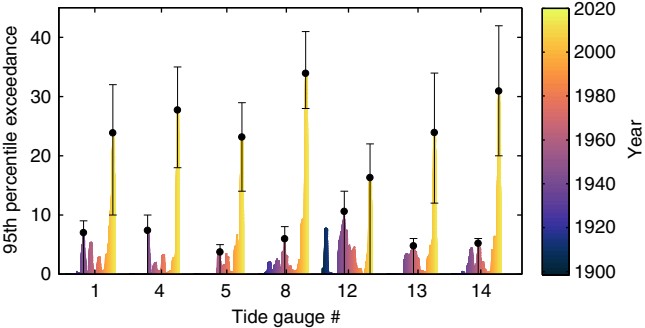

**Fig. 2** Exceedances of the 95th percentile. Number of months within 7-year running windows for which the amplitude of the SLAC is above the 95th percentile (computed over the entire record) for tide-gauge records with at least 50 years of data. To construct the histogram, a 7-year window is shifted month by month starting with a window centered at month 43 of the record. Numbers along the *x* axis refer to the identification numbers shown in Fig. 1, while colors correspond to the color bar and denote time. The two error bars in each histogram represent the 68% credible intervals associated with the maximum values in 2008–2009 and in the period before 1990, whereas the black dots represent such maximum values

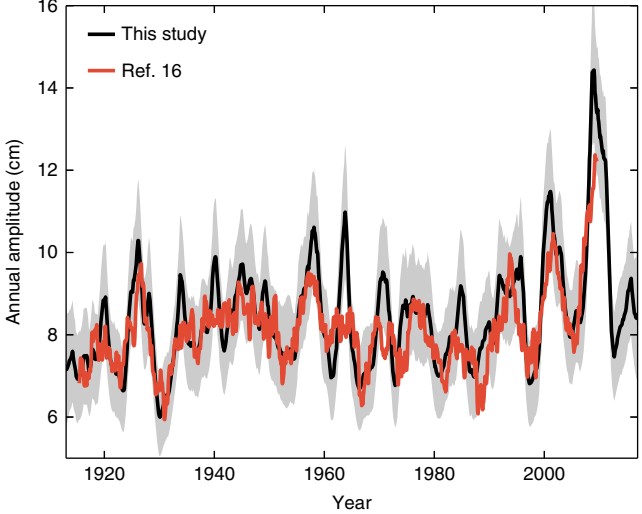

**Fig. 3** SLAC amplitude at Key West. Comparison of the SLAC amplitude for the Key West tide gauge as estimated with a Bayesian state-space model (black) and by ref. [16] using the method of 5-year running windows (red). The gray-shaded area represents the 68% (1-sigma) credible interval for our estimate

different sides of the Cape. The existence of two different regimes north and south of Cape Hatteras has been observed previously for inter-annual sea-level variability[24,25]. Given the larger amplitude variations and the high vulnerability of the Gulf and Southeast coasts to sea-level changes, hereafter we focus on this region.

A prominent feature of the time-varying amplitudes is their particularly large values around 2009 uniformly across all stations south of Cape Hatteras. This feature is further emphasized by plotting the number of months for which the annual amplitude is above the 95th percentile in 7-year running windows for the longest tide-gauge records (Fig. 2). The maximum number of exceedances is found in 2008–2009 at all stations and is likely (probability ≥68%) unprecedented in the tide-gauge record as indicated by the non-overlapping credible intervals. An amplification of the SLAC after 1990 was reported recently for stations in the Gulf of Mexico[16], but it was not clear from that study whether and to what extent changes after 1990 represented a sustained change. Here we clarify this issue and show that such changes do not represent a permanent amplification of the SLAC but consist of a succession of decadal fluctuations with a particularly large peak around 2009. We illustrate this by plotting the amplitude at Key West as derived from our method together with the estimate of ref. [16] based on the windowing method (Fig. 3). Overall, the two time series are in good agreement, though the latter shows reduced fluctuations and misses some features such as the peaks in the 1960s and the early 1970s. Importantly, the last value in the estimate of ref. [16] is for June 2009, which coincides exactly with the time of the highest peak over the entire record. This coincidence results in a curve that is characterized by a relatively flat period until 1990 and a marked rise from that point onwards, giving the impression of a sustained change. Our estimate, however, shows that the annual amplitude fell back to average values after 2009 as part of a large decadal oscillation (Fig. 3), limiting support for the existence of a long-term trend but revealing the presence of an enhanced fluctuation at the end of the record.

**Mechanisms of changes in the annual amplitude.** The coherent signal observed by tide gauges could represent either a coastal signal or a basin-scale mode where both the coastal zone and the

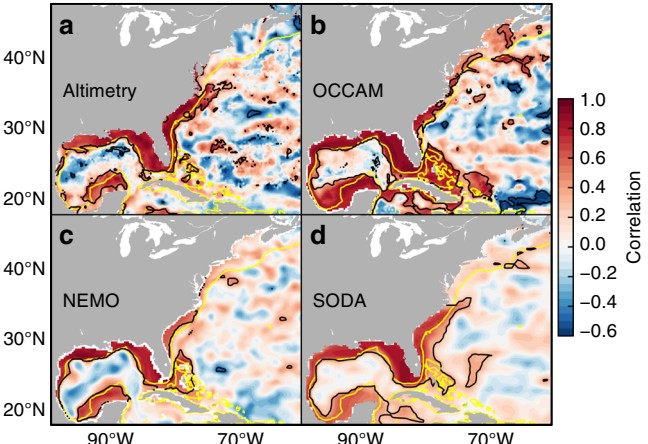

**Fig. 4** Correlation maps from satellite altimetry and ocean models. Point-wise correlation between the amplitude of the SLAC at each grid point and that averaged along the United States Gulf and Southeast coasts for **a** altimetry (1993–2016), **b** OCCAM (1985–2003), **c** NEMO (1968–2012), and **d** SODA (1900–2010). The average has been computed over grid points within the 0–500 m depth range following the coast from Pensacola to Charleston. Black line denotes significance of the correlation at the 95% confidence level, whereas yellow line represents the 500 m isobaths

deep ocean oscillate together. Determining which of the two cases applies is crucial to understanding the true nature of this signal, but such determination cannot be made solely on the basis of tide gauges located on the coast. To shed light on this issue, we have computed the point-wise correlation between the annual amplitude from satellite altimetry data at each grid point and that averaged along the United States Gulf and Southeast coasts (Fig. 4a). The correlation map shows that changes in the amplitude are coherent along the coast from the Yucatan Peninsula to Cape Hatteras. However, the coherence is confined to the coastal zone. The altimetry data covers only the period 1993–2016, therefore the question arises as to whether the correlation pattern

depends on the period considered or its length. To address this question we have computed analogous maps based on data from the Ocean Circulation and Climate Advanced Modelling (OCCAM) project model (Fig. 4b), the Nucleus for European Modelling of the Ocean (NEMO) model (Fig. 4c), and the Simple Ocean Data Assimilation (SODA) reanalysis (Fig. 4d) (see Methods for details of the models). The three model-based patterns are very similar to that from altimetry, showing strong coherence along the coast south of Cape Hatteras and providing confidence in the robustness of the correlation spatial structures.

Sea-level changes can be partitioned into the sum of three components: steric, mass, and the inverse barometer (IB) effect (Methods). Different processes contribute differently to these components, and thus establishing the dominant component generally reveals key information on the underlying mechanisms. To this end, we have computed the correlation of the steric

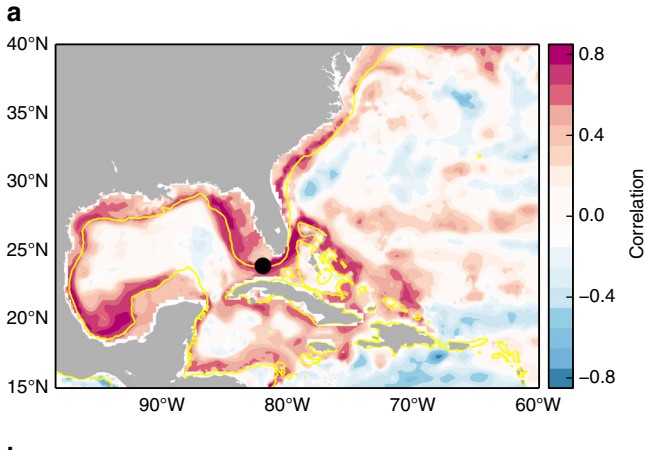

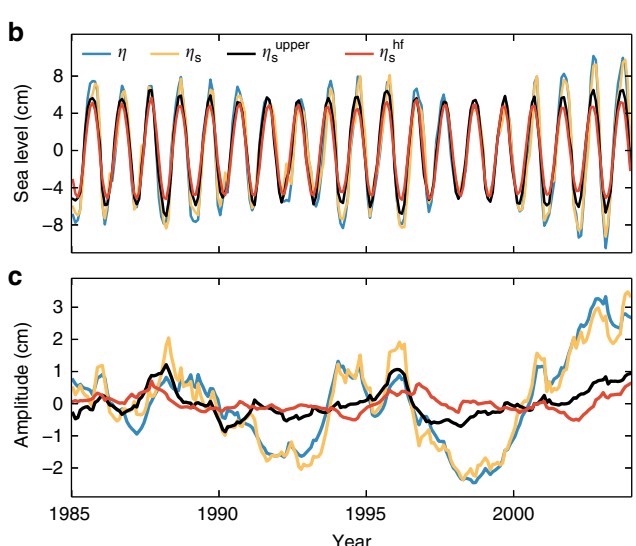

**Fig. 5** Correlation map and time series of the steric component from OCCAM. **a** Point-wise correlation of the steric annual amplitude at each grid point with the SLAC amplitude averaged along the United States Gulf and Southeast coasts. The average has been computed over grid points within the 0–500 m depth range following the coast from Pensacola to Charleston. The yellow line represents the 300 m isobath. **b** The annual cycle of total sea level (blue), total steric height (orange), and the steric contributions from above the seasonal thermocline (black) and due to surface heat fluxes (red) at the location denoted by the black dot shown in **a**. **c** The annual amplitude (time mean removed) of the time series shown in **b**

annual amplitude at each grid point with the amplitude of the SLAC averaged along the Gulf and Southeast coasts in the OCCAM model (Fig. 5a). The highest correlations are found predominantly along the continental slope, suggesting that coastal changes in the annual amplitude are attributable to steric changes. The coherence of the steric signal stretches from the Yucatan Peninsula to Cape Hatteras following the slope. The low correlations at the coast are explained by the fact that the steric component is defined as a depth integral and, hence, is necessarily small in shallow waters. Note, however, that steric signals over the slope can be transmitted to the coast through an indirect effect on bottom pressure.

We further confirm the dominant role of the steric component by analyzing time series of the SLAC amplitude from OCCAM along the Gulf and Southeast coasts. In particular, we show that while the mean SLAC is primarily driven by the expansion and contraction of the water column above the seasonal thermocline (top ~70 m) due to changes in surface heat fluxes, the modulation of the SLAC is due to steric changes in deeper layers. The contribution of surface heat fluxes to the SLAC, $\eta_s^{hf}$, is estimated using Eq. (4), and is compared to the steric contribution from above the seasonal thermocline, $\eta_s^{upper}$, as given by Eq. (2), as well as to the total steric and the total sea level. The results of the analysis for an arbitrary location in the Gulf of Mexico are shown in Fig. 5. We find that $\eta_s^{hf}$ explains 95% of the variance in the annual cycle of $\eta_s^{upper}$, and in turn the latter explains 89% of the variance in the SLAC (Fig. 5b). However, neither $\eta_s^{hf}$ nor $\eta_s^{upper}$ explains very much of the changes in the amplitude of the SLAC (Fig. 5c). In contrast, the total steric explains the majority (91%) of the changes in the amplitude of the SLAC. Similar results are found at other locations along the slope. Two implications can be drawn. First, all the information on the modulation of the SLAC resides in the ocean layers below the seasonal thermocline. Second, the SLAC can be simply described as the sum of the unmodulated cycle and a term representing steric changes below the seasonal thermocline (hereafter referred to as the modulator):

$$\text{SLAC} = \underbrace{\text{SLAC}_{\text{mean}}}_{\substack{\text{Steric above seasonal thermocline}}} + \underbrace{\text{SLAC}_{\text{modulator}}}_{\substack{\text{Steric below seasonal thermocline}}} .$$

These two implications affect how we understand the SLAC modulation and provide the basis for the subsequent analysis. In this regard, note the following. In the time domain, amplitude modulation typically involves multiplication of a low-frequency modulating signal and a high-frequency sine wave (the latter is often termed the carrier in radio communications). However, from properties of the Fourier transform, multiplication in the time domain corresponds to convolution in the frequency domain. Therefore, in the frequency domain, amplitude modulation appears as sums and differences of the frequencies of the two input signals. This implies that any modulated signal can always be mathematically described as the sum of the carrier and a superposition of sinusoids with frequencies slightly above and below the carrier frequency (see Methods for proof). This alternative interpretation is exactly analogous to the steric representation of the SLAC modulation. It turns out that the ocean, along its vertical dimension, behaves similarly to a Fourier transform in that it separates the frequency components of the SLAC into different ocean layers. This result will greatly facilitate our analysis.

The fact that changes along the coast are correlated over large distances but are decoupled from nearby deep-ocean changes is highly suggestive of fast wave propagation along the coast and indicates that local forcing is an unlikely driving factor. Indeed, the local response to changes in atmospheric pressure, quantified

through Eq. (1) (Methods), explains <5% of the variance in the annual amplitude at all tide gauges. Similarly, we find no statistically significant correlation with local wind changes at any station. A number of mechanisms may be invoked to explain the correlation patterns (Figs. 4 and 5). The first one involves the generation of coastally trapped waves[26,27] by longshore wind or buoyancy forcing (e.g., a river). These waves propagate along the boundary with the coast on the right (in the Northern Hemisphere) at speeds of a few m/s (first baroclinic mode), have an offshore length scale of about 50 km, and can carry the effects of the forcing over large distances along the coast. Importantly, the thermocline displacements associated with these waves are correlated with sea-level changes at the coast, and thus are captured by tide gauges. Propagation of sea-level anomalies along the coast has been observed in many regions[28–30]. The second plausible mechanism involves the generation of boundary waves by incident Rossby waves from the ocean interior[31], which could, similarly, affect coastal sea level over large sections of coastline. Processes of Rossby wave generation include wind-stress-curl[32] and buoyancy[33] forcing. In the following, we explore which of these two mechanisms is more likely to explain the observed changes in the SLAC amplitude.

We have assessed the role of longshore wind by means of the model described in Appendix A of ref. [34]. In particular, we have integrated the model equation from north to south starting at Cape Hatteras using a range of values for the length decay scale (100–1000 km), but have found no agreement with the changes in the amplitude of the SLAC from tide-gauge records. In addition, we have compared the SLAC from tide gauges with the annual cycle of river discharge for the major rivers in the United States flowing into the Atlantic, again without finding a good agreement. This leaves us with the incidence of Rossby waves on the western boundary as the most likely mechanism. In the following, we concentrate on this possibility and explore it on the basis of the OCCAM and NEMO models.

To investigate the role of Rossby waves in controlling the SLAC modulation, we focus on the region east of the Bahamas. The reason for this choice is that Rossby waves play a particularly important role in driving sea-level variability in this region[32]. In addition, changes in the SLAC amplitude in this region are significantly correlated with changes along the coastline of the mainland United States (Figs. 4 and 5), which suggests a common driving mechanism. This location is also convenient because at this latitude the Gulf Stream is restricted to the Florida Strait and hence does not interfere with the Rossby waves reaching the Bahamas east coast.

We have computed the correlation at different lags of the steric contribution from below the seasonal thermocline at the continental slope east of the Bahamas with that at each grid point in both OCCAM and NEMO. For grid points in shallow areas (<200 m), the correlation is computed with the SLAC modulator instead of the steric. The pattern of evolution (Fig. 6) shows a region of significant correlation several hundred kilometers off the coast of the Bahamas at lags of ~3 months, indicating a lagged relationship between this region and the western boundary. As the lag decreases, the region of correlation propagates westward until it reaches the coast at lag zero and then the entire shelf and coastal zone become significantly correlated, both in the Gulf of Mexico and along the Southeast coast. The close resemblance between the maps from the two models provides confidence in the robustness of this spatiotemporal pattern. We conclude that the SLAC modulator along the Gulf and Southeast coasts is related to density anomalies below the seasonal thermocline propagating westward.

Further supporting evidence for the link to propagating anomalies is provided by producing a time-longitude section of the steric modulator east of the Bahamas at 26.5°N in OCCAM (Fig. 7a). The SLAC modulator along the Gulf and Southeast coasts is consistent with steric anomalies that originated in the ocean interior at earlier times, as indicated by the alignment of peaks and troughs in the time series of the SLAC modulator and the steric modulator at the Bahamas coast. While often the steric anomalies are formed far in the interior of the Atlantic, sometimes they originate only a few hundred kilometers offshore and reach the coast after a few months. Our calculations show that the density anomalies propagate at an average speed of about 4.1 cm/s, which is consistent with the observed phase speed of long Rossby waves at this latitude[35].

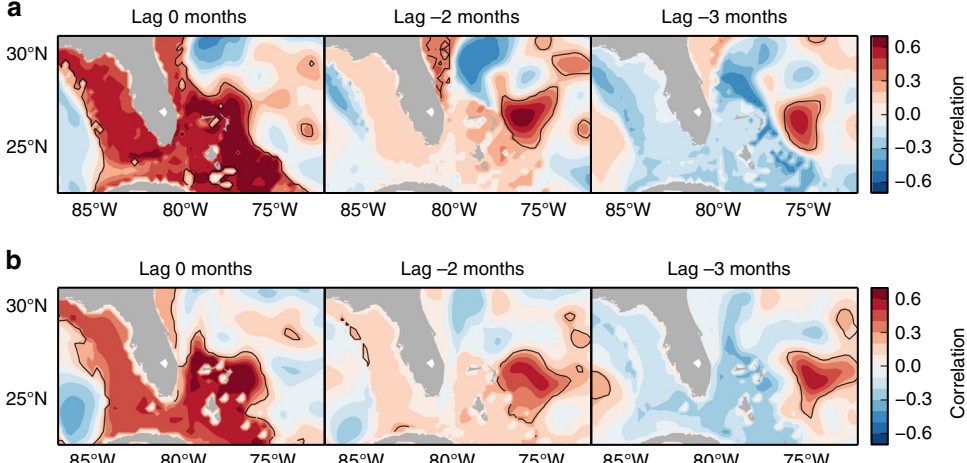

**Fig. 6** Lagged correlation maps of the steric contribution from below the seasonal thermocline. Point-wise lagged correlation of the steric contribution from 200 to 1000 m depth at the continental slope east of the Bahamas with that at each grid point for **a** OCCAM and **b** NEMO. Steric time series have been band-pass filtered (Butterworth filter with lower and higher cutoff frequencies: 1/16 and 1/8 months⁻¹) to focus on the frequencies relevant to the SLAC modulator. For grid points in shallow areas (<200 m), the correlation is computed with the SLAC modulator instead of the steric. Negative lags indicate that steric changes at the slope east of the Bahamas lag relative to other grid points. Black line denotes significance of positive correlations at the 95% confidence level

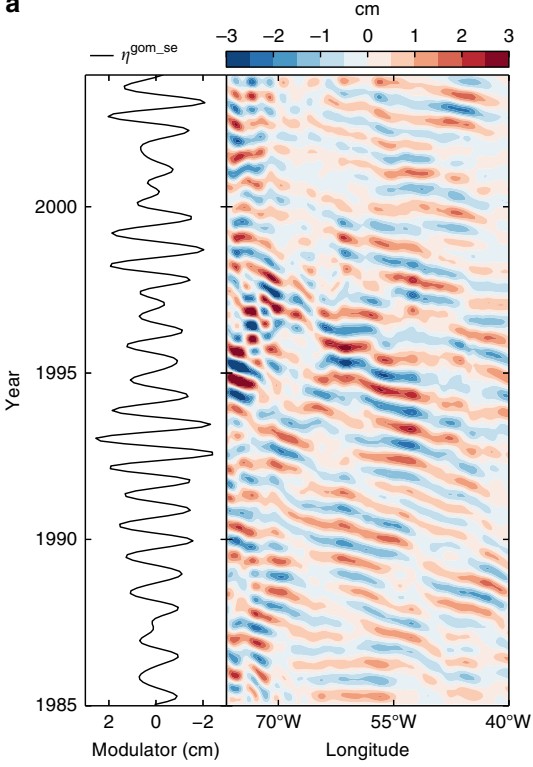

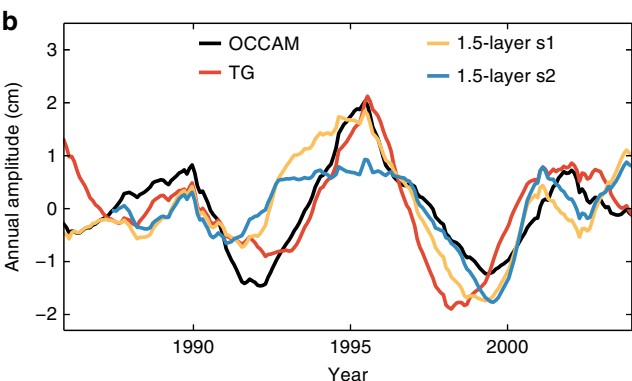

**Fig. 7** Hovmöller diagram of the steric modulator and estimates from a reduced gravity model. **a** Time-longitude section of the steric modulator for grid points east of the Bahamas at 26.5°N in the OCCAM model. The time series of the SLAC modulator averaged along the United States Gulf and Southeast coasts ($\eta^{\text{gom-se}}$) is also shown to the left. **b** Estimates of the SLAC amplitude at the Bahamas east coast based on a 1.5-layer, reduced gravity model for solutions one (orange) and two (blue), along with the SLAC amplitude (time mean removed) from OCCAM at the Bahamas east coast (black) and that averaged over tide-gauge stations 1–7 (red)

The results above suggest that a simple model based on Rossby wave dynamics might be used to capture the modulation of the SLAC along the Gulf and Southeast coasts. To test this, we use a 1.5-layer, reduced gravity model forced by wind (Methods). We compute two solutions. In the first solution, we start the integration at $x_e = 66.5°W$ (~1000 km from the Bahamas coast) and set the value of $\eta$ at $x_e$ equal to the OCCAM sea level, while in the second one, we set $x_e = 46.5°W$ (~3000 km from the Bahamas coast) and $\eta$ at $x_e$ to zero. The first solution includes the effects of the wind between the Bahamas and $x_e$ plus any contribution originating to the east of $x_e$ (wind-driven or otherwise), while the second solution includes only the effects of the wind. Starting the

integration further to the east in the second solution changes the results only marginally. The reduced gravity model gives a good match to both tide-gauge observations and OCCAM data (Fig. 7b), providing strong evidence for a physical link between the SLAC and Rossby waves. In particular, the correlation between the modeled and observed SLAC amplitude is 0.67 and 0.65 for the first and second solutions, respectively. These values are comparable to the correlation of the OCCAM estimates with tide-gauge data (0.7). The strong resemblance between the two solutions along with the good agreement with observations indicate that wind forcing is a dominant cause of the Rossby waves. We note, however, that the second solution slightly underestimates the peak in 1995 relative to the first solution, suggesting that other drivers and/or non-linear effects may also play a role.

It is interesting to assess whether the incident density anomalies are modified by the sloping topography when they approach the western boundary. To this end, we have computed the standard deviation of the SLAC modulator as a function of distance from the Bahamas coast along with the correlation between the modulator at the coast and that offshore (Supplementary Fig. 2). While there is a gradual decrease in the magnitude of the modulator with proximity to the boundary, the phase coherence remains significant through the continental slope as indicated by the correlation between the modulator at the coast and that in the open ocean. The reduction in dynamic height variability toward the western boundary has been reported before and is explained by frictional energy dissipation and the export of energy through boundary waves[31,36,37]. The latter is precisely the mechanism that we invoke to explain the coherence of the amplitude over large distances along the coast.

It is also interesting to note that the meridional coherence scale of the westward-propagating density anomalies is relatively small (Fig. 6). Nevertheless, both observations and models show that changes in the amplitude of the SLAC are coherent along the entire coastline up to Cape Hatteras. Because boundary waves propagate along the coast with the coast to the right, the coherence at latitudes north of the Bahamas may suggest an effect of the Rossby waves on the Gulf Stream. This would be consistent with results from previous studies that showed a significant response of the Gulf Stream to incident density anomalies from the ocean interior[38,39]. In particular, it has been found that, on the timescales relevant to the SLAC modulator (~annual), the Florida Current responds almost instantaneously to incident density anomalies just east of the Bahamas leading to a significant anti-correlation with the UMO. This response of the Florida Current could explain the coherence of the SLAC amplitude at high latitudes. In support of this premise, we find that the SLAC modulator from tide gauges along the Southeast coast (stations 10–14) is correlated (−0.36, significant at the 95% confidence level) with band-pass filtered (1/20–1/5 months$^{-1}$) variations of the Florida Current transport.

In summary, we have shown that the mean SLAC is driven by steric changes above the seasonal thermocline induced by variations in surface heat fluxes, while the SLAC modulation is related to density changes between 200 and 1000 m depth that originate in the ocean interior and propagate westward as Rossby waves. Upon impinging on the western boundary, we conjecture that the Rossby waves generate boundary waves that propagate rapidly along the continental slope giving rise to highly-coherent sea-level changes along the coast. A schematic illustration explaining the proposed mechanisms is shown in Fig. 8. It should be noted that our results regarding the variability associated with the modulator are general in that they do not depend on whether the annual cycle is interpreted as a changing or repeating cycle. By definition, the modulator is closely related

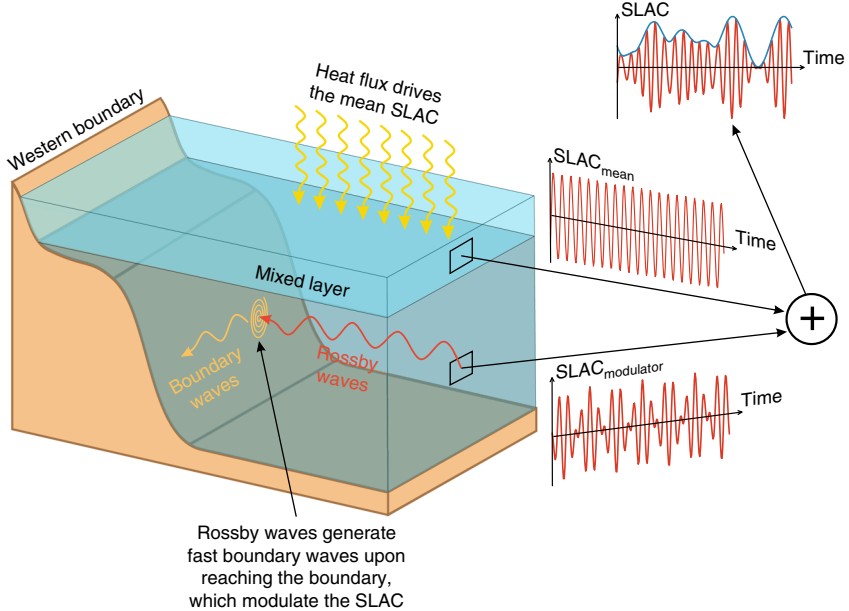

**Fig. 8** Schematic illustration of the mechanism of SLAC modulation. The mean SLAC is associated with steric changes in the seasonal thermocline induced by variations in surface heat fluxes, whereas its modulation is related to density anomalies in deeper layers propagating westward as Rossby waves. These Rossby waves give rise to fast boundary waves upon impinging on the western boundary, which in turn modulate the SLAC along the Gulf and Southeast coasts and lead to the coherence over large distances along the coast

to the variability that results from removing a stationary annual cycle and then applying a band-pass filter around the relevant frequencies. This implies that approaches that assume a stationary annual cycle and focus on the frequencies of the modulator will reach the same conclusions as presented here, with the difference that such approaches will not interpret the variability as being part of a modulated annual cycle but rather as anomalies relative to a repeating cycle.

Finally, it must be noted that there is a theoretical upper limit on the frequency of Rossby waves beyond which such waves cannot exist. This limit follows from the dispersion relation and for long Rossby waves varies with latitude according to[19] $\omega_{max} = (c/2R)\cot\varphi$, where $c$ is the baroclinic gravity-wave phase speed, $R$ is the radius of the earth, and $\varphi$ denotes latitude. The dependence on latitude imposes a constraint on where Rossby waves might act as the SLAC modulator because this possibility requires waves with nearly annual periods. In particular, spectral analysis of the modulator reveals an upper sideband of ~10.5 months at all tide-gauge stations (Supplementary Fig. 3). It follows then that Rossby waves with periods of ~10.5 months are required, but these are only possible at latitudes south of ~40°N.

**The relationship between the SLAC and the UMO transport.** Previous studies[37,40] have found that Rossby waves and oceanic eddies impinging on the western boundary can have a significant effect on the geostrophic component of the MOC, especially at intra-annual timescales. Therefore, a question arises as to whether the propagating density anomalies responsible for the changes in the SLAC amplitude exhibit themselves also in this component of the MOC. To explore this possibility, we analyze time series of UMO transport from RAPID[23] for the period April 2004 to October 2015 and from OCCAM for the period 1985–2003. The UMO transport is related to the horizontal difference in pressure between the eastern and western boundaries. If a relationship exists between the SLAC and the UMO, this is most likely due to

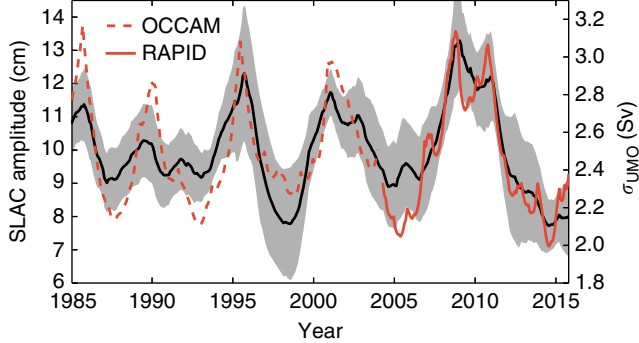

**Fig. 9** SLAC from tide-gauge records and the UMO transport. Instantaneous amplitude of the SLAC (left axis) averaged over tide-gauge stations 1–10 together with the instantaneous standard deviation of the UMO transport at 26.5°N (right axis) both from RAPID (red solid) and the OCCAM model (red dashed). The gray-shaded area denotes the standard deviation about the average annual amplitude of the 10 tide-gauge stations. A long-term trend has been removed from all time series

the western-boundary contribution to the UMO, where the influence of incident Rossby waves occurs[37,40]. Such contribution, however, does not have an annual cycle but instead exhibits non-periodic variations. For non-periodic signals, the notion of peak amplitude is not well defined, so to relate the UMO to the SLAC amplitude we use the instantaneous variance of the UMO transport as a measure of its amplitude or intensity. We expect changes in the UMO variance (at the frequencies of the SLAC modulator) to covary with changes in the SLAC amplitude. To estimate the instantaneous variance of the UMO transport at the relevant timescales, we use a stochastic variance model (see Methods for details). We find a strong relationship between the amplitude of the SLAC and the variance of the UMO transport, wherein larger annual amplitudes are associated with increased

fluctuation intensity in the UMO (Fig. 9). The correlation between the two quantities is higher for RAPID observations (0.91) than for OCCAM data (0.75), but in both cases it is significant at the 95% confidence level. The implication is that the density anomalies that modulate the SLAC also affect the UMO transport by altering the zonal pressure gradient through density variations at the western boundary.

Our analysis of tide-gauge records has revealed the presence of large inter-annual to decadal variations in the amplitude of the SLAC along the United States Gulf and Southeast coasts, which have been particularly large since the 1990s. Because the SLAC in this region peaks during the period of maximum hurricane activity in the Atlantic (between August and October), larger annual amplitudes imply increased risk of damage from hurricane storm surges due to a higher base water level. In addition, larger seasonal variations also significantly increase the likelihood of nuisance flooding and exacerbate other direct effects of annual sea-level changes (e.g., erosion, estuary productivity, and so on). Furthermore, the variations in the amplitude of the SLAC are coherent over large distances along the coast, which means that the increased risk associated with them is not localized but affects the entire coastline at any particular time. Importantly, we have suggested that these variations are associated with incident baroclinic Rossby waves from the open ocean. Since these waves propagate slowly, their effects on the coastal SLAC are felt months or even years after they are formed in the ocean interior. This delayed coastal response raises the possibility of seasonal forecasts of water levels in coastal areas, which would allow coastal managers and communities to better assess and mitigate associated risks. We have also provided observational and model-based evidence of a link between the SLAC amplitude and the UMO transport, wherein larger SLAC amplitude coincide with amplified annual UMO variations. This result suggests that long tide-gauge records could be used to infer properties of the UMO variability for periods during which no direct estimates are available (i.e., before 2004). Given the role of the MOC in northward heat transport and the climate, the new link between the SLAC and the UMO is of particular importance and will aid current efforts to better understand the behavior of this key component of the climate system.

## Methods

**Data sets and ocean models**. Monthly mean values of sea level from tide-gauge records are obtained from the Revised Local Reference data archive of the Permanent Service for Mean Sea Level[41] (http://www.psmsl.org/). The location of the tide gauges used in this study is shown in Supplementary Fig. 1. Sea-level pressure and wind monthly data are obtained from the 20th Century Reanalysis v2c30[42] for the period before 2015 and from the National Centers for Environmental Prediction reanalysis[43] for the period 2015–2016 (both data sets are available at http://www.esrl.noaa.gov/psd/data). Monthly flow rates of rivers in the United States flowing into the North Atlantic are from the Research Data Archive at the National Center for Atmospheric Research (https://rda.ucar.edu). Monthly values of net surface heat flux covering the period 1983–2009 were provided by the WHOI OAFlux project (http://oaflux.whoi.edu). The time series of the UMO transport is obtained from the RAPID-WATCH MOC monitoring project[44] (available at http://www.rapid.ac.uk), which is provided at twice daily resolution and covers the period from April 2004 to October 2015. Daily mean transport estimates of the Florida Current from a submarine cable and calibration cruises covering the period 1982–2016 were obtained from the Atlantic Oceanographic and Meteorological Laboratory web page (www.aoml.noaa.gov/phod/floridacurrent/). Both the UMO and the Florida Current transport time series are averaged into monthly values.

The satellite altimetry data are obtained from the multi-mission gridded sea surface heights product provided by the Copernicus Marine Environment Monitoring Service (available at http://marine.copernicus.eu/). The data are made available as weekly fields on a 1/4° × 1/4° near global grid covering the period from January 1993 to May 2016. These weekly fields are averaged into monthly fields for our analysis. The data are provided with all standard corrections applied, including corrections for tropospheric (wet and dry) and ionospheric path delays, sea state bias, tides (solid earth, ocean, loading, and pole), and atmospheric effects (sea-level pressure and high-frequency winds).

In this study, we use data from the OCCAM model. The version that we use here is a free-surface free-running (without data assimilation) global model with a spatial resolution of 1/4° × 1/4° in the horizontal and 66 non-uniform z-levels in the vertical, covering the period 1985–2003[45]. We also use data from the NEMO (1/4°) global ocean model in its ORCA025[46] (available from http://www.ceda.ac.uk/projects/jasmin), which covers the period 1958–2012 and from the SODA reanalysis[47], which covers the period 1871–2010. The NEMO model is not spun up prior to 1958, and thus to make sure that we start from a stable ocean state we consider only NEMO data from 1968.

As a validation, we have compared annual amplitudes derived from OCCAM and NEMO with those from tide-gauge observations (Supplementary Fig. 4). Both models show significant correlations at most tide-gauge stations, though OCCAM performs better than NEMO as indicated by the higher correlations. It should also be noted that the correlation maps from OCCAM and NEMO are very similar between them and also to altimetry (Fig. 4). The good agreement between model data and observations gives us confidence in the capability of the two models to capture the dynamics governing the observed changes in the SLAC.

**Sea-level equations**. It is convenient for our purposes to describe sea-level changes, $\eta$, as the sum of three components: (1) the IB effect, $\eta_{IB}$, representing the effect of changes in sea-level pressure; (2) the steric component, $\eta_s$, representing the effect of variations in the ocean density field; and (3) the mass component, $\eta_m$, representing the effect of mass redistribution within the Earth system unrelated to changes in sea-level pressure. Expressions for each of these components are obtained from integration of the hydrostatic relation[7]:

$$\eta_{IB} = \frac{1}{g\rho_0}\left(\overline{P_a} - P_a\right), \quad (1)$$

$$\eta_s = -\frac{1}{\rho_0}\int_{-H}^{0}\rho\, dz, \quad (2)$$

$$\eta_m = \frac{1}{g\rho_0}\left(P_b - \overline{P_a}\right), \quad (3)$$

where $g$ is the gravitational acceleration, $\rho_0$ is a reference density (1025 kg m$^{-3}$), $P_a$ is the atmospheric sea-level pressure anomaly, $\rho$ is the in situ density anomaly of the water, $P_b$ is the pressure anomaly at the ocean bottom $z = -H$, and the overbar denotes spatial average over the global oceans. Note that, as written, Eq. (2) gives the total steric contribution, but the same equation can also be used to calculate the steric contribution, for example, from above the seasonal thermocline simply by replacing $H$ with the appropriate depth. The reference depth that we use to compute the steric contribution from above the seasonal thermocline is 70 m, which corresponds to the average depth of the mixed layer in OCCAM. Selecting other values in the range 60–120 m did not affect our results in any significant manner.

The steric contribution due to changes in surface heat fluxes, $\eta_s^{hf}$, can be estimated from the following first-order linear equation[7]:

$$\frac{\partial\eta_s^{hf}}{\partial t} = \frac{\alpha}{\rho_0 C_p}\left(Q_{net}(t) - \langle Q_{net}(t)\rangle\right), \quad (4)$$

where $\alpha$ is the coefficient of thermal expansion, $C_p$ is the specific heat of sea water, $Q_{net}$ is the net surface heat flux, and the angle brackets denote temporal averaging. $\alpha$ is estimated from the OCCAM temperature and salinity fields averaged over the mixed layer. The depth of the mixed layer is determined using a potential density threshold of 0.125[48] (sigma units) relative to the density at the first model level (2.7 m).

**The 1.5-layer reduced gravity model**. To quantify the contribution of Rossby waves to the modulation of the SLAC, we use a 1.5-layer, reduced gravity model forced by wind stress. Under the long-wave and quasi-geostrophic approximations, the equation describing the evolution of sea level, $\eta$, can be written as[49]

$$\frac{\partial\eta}{\partial t} - C_R\frac{\partial\eta}{\partial x} + R\eta = -\frac{g'}{g}\mathbf{k}\cdot\nabla\times\left(\frac{\boldsymbol{\tau}}{\rho_0 f}\right), \quad (5)$$

where $\boldsymbol{\tau}$ is the wind-stress vector, $f$ is the Coriolis parameter (allowed to vary with latitude), $g'$ is the reduced gravity, $R$ is the decay rate, $C_R$ is the propagation speed of long baroclinic Rossby waves, and $\mathbf{k}$ is the vertical unit vector. Here we choose $C_R = 4$ cm s$^{-1}$, $R = (1.5$ years$)^{-1}$, and $g' = 3$ cm s$^{-2}$. Our results are fairly insensitive to the choice of the model parameters within the typical ranges $(1$ year$)^{-1} \leq R \leq (2$ years$)^{-1}$ and 2 cm s$^{-2} \leq g' \leq 4$ cm s$^{-2}$. We want to calculate $\eta$ at 26.5°N on the east coast of the Bahamas. This is done by integrating Eq. (5) from a point to the east of the Bahamas, $x_e$, along the

baroclinic Rossby wave characteristic

$$\eta(x_B, t) = \eta\left(x_e, t + \frac{x_B - x_e}{C_R}\right) \exp[R(x_B - x_e)/C_R]$$
$$+ \frac{g'}{gC_R} \int_{x_e}^{x_B} \mathbf{k} \cdot \nabla \times \left[\tau\left(x', y, t + \frac{x_B - x'}{C_R}\right)/(\rho_0 f)\right] \exp[R(x_B - x')/C_R] dx' \tag{6}$$

where $x_B$ is the point at which the solution is wanted (i.e., the Bahamas east coast).

Solutions are based on the same wind-stress data that was used to force OCCAM, which allows us to evaluate the performance of the model by comparison with OCCAM. To focus on the frequencies relevant to the SLAC modulator, a Butterworth band-pass filter (lower and higher cutoff frequencies: 1/16 and 1/8 months$^{-1}$) is applied to the solution.

**Calculation of the upper mid-ocean transport**. Following ref. [50], the UMO transport from OCCAM at 26.5°N has been obtained by first computing the zonally integrated northward geostrophic transport per unit depth, $T(z)$, as

$$T(z) = \frac{P_E(z) - P_W(z)}{\rho_0 f}, \tag{7}$$

where $P_E(z)$ and $P_W(z)$ denote pressure at the eastern and western boundaries of the North Atlantic, respectively, and $f$ is the Coriolis parameter. $P_E(z)$ is calculated at the easternmost grid point for any given depth, whereas $P_W(z)$ is calculated at the westernmost grid point where water depth is at least 4800 m (i.e., a vertical profile ~25 km from the coast). The meridional transport due to the flow between the western vertical profile and the Bahamas east coast (referred to as the western-boundary wedge), $T_{WBW}(z)$, is estimated directly from the model velocities. The UMO transport is then given by the depth integral of $T(z) + T_{WBW}(z)$ between the surface and 1100 m:

$$\text{UMO} = \int_{-1100}^{0} [T(z) + T_{WBW}(z)] dz. \tag{8}$$

**State-space model for the annual cycle**. To estimate the instantaneous amplitude and phase of the SLAC, we use a state-space model. The state-space approach provides a powerful framework for addressing problems such as the one at hand in which we wish to estimate, based on indirect information from noisy observations, a set of state variables (e.g., the amplitude of the annual cycle) that are not directly measurable. Here we formulate the inference problem in terms of a non-linear state-space model and adopt a Bayesian approach, thus modeling the unknown static parameters of the model as random variables. For the representation of a system in state-space form, two types of equations are required[22]. In standard state-space notation, let $y_{1:T} = (y_1, ..., y_T)$ be a sequence of observations (e.g., a tide-gauge record), $\mathbf{x}_t \in \mathbb{R}^n$ denote the latent state at time $t$, and $\theta$ denote the unknown static parameters of the model. The state-space model then consists of the transition probability density $p_\theta(\mathbf{x}_t|\mathbf{x}_{t-1})$ describing the evolution of the state variables with time, and a measurement model linking the observations to the state as defined by the probability density $p_\theta(y_t|\mathbf{x}_t)$. Our goal is to compute the joint state and parameter posterior distribution given all observations $p(\theta, \mathbf{x}_{1:T}|y_{1:T})$.

One key advantage of our method is that it computes estimates conditioned on the full history of sea-level observations, which significantly improves the resolvability of the state variables. In other words, the method uses all available past and future observations to estimate the state of the system at each time step. In addition, the method provides estimates for the entire period covered by the sea-level observations, including the edges of the time series. These are important distinctions to the method based on a harmonic fit to running windows. Another key aspect of our method is that it allows for parameter uncertainty and involves rigorous error propagation, thus providing realistic uncertainty estimates. Furthermore, the method does not rely on large data samples to be accurate, and is relatively insensitive to starting values in the parameters. One limitation of the method is that it is computationally expensive.

Here the sea-level time series are modeled as the sum of four terms: (1) an annual cycle with instantaneous amplitude $a_t^a$ and phase $\phi_t^a$; (2) a semi-annual cycle with instantaneous amplitude $a_t^{sa}$ and phase $\phi_t^{sa}$; (3) a low-frequency component $b_t$, which includes any existing non-linear trend; and (4) white Gaussian noise $e_t$. The measurement model takes the following form:

$$y_t = a_t^a \cos\phi_t^a + a_t^{sa} \cos\phi_t^{sa} + b_t + e_t, \quad e_t \sim \mathcal{N}(0, \sigma_0^2), \tag{9}$$

where $\mathcal{N}(m, \sigma^2)$ denotes the normal distribution of mean $m$ and variance $\sigma^2$, and $\sigma_0^2$ is a parameter to be estimated (and thus contained in $\theta$).

Our state-space model has been designed to incorporate realistic dynamics for all state variables while at the same time keeping it simple enough to make Bayesian inference feasible. In this regard, two aspects of the state dynamics in particular merit careful consideration when designing the state transition kernel. First, the frequency of either the annual or semi-annual cycles may change over time but it should not drift too far away from its mean value. In other words, the frequency of the cycles should be stationary, but not an iid process since the

frequency should be allowed to deviate from its mean value for certain periods of time. This is achieved by modeling $\phi_t^a$ as an integrated process of order one with the phase increments $\omega_t^a = \phi_t^a - \phi_{t-1}^a$ following a first-order autoregressive (AR1) process (and similarly for the phase of the semi-annual cycle $\phi_t^{sa}$).

The second aspect that requires consideration concerns the fact that the amplitude is a non-negative-valued variable. To satisfy this requirement, we model the logarithm of the amplitude of the annual and semi-annual cycles ($\lambda_t^a$ and $\lambda_t^{sa}$, respectively), as a random walk, i.e. $p_\theta(\lambda_t^a|\lambda_{t-1}^a) = \mathcal{N}(\lambda_t^a; \lambda_{t-1}^a, \sigma_1^2)$, where $\sigma_1^2$ is a parameter to be estimated. The amplitude is then obtained by taking the exponential of the log amplitude, i.e., $a_t^a = \exp\lambda_t^a$. This approach is standard in Bayesian statistics and has the great advantage of allowing us to place a conjugate prior on the unknown parameter $\sigma_1^2$ resulting in a closed-form expression for the posterior distribution, thus greatly facilitating the task of sampling from such distribution.

The evolution of the state variables is modeled as follows.
Log amplitude of the annual and semi-annual cycles:

$$\lambda_t^a = \lambda_{t-1}^a + q_t, \quad q_t \sim \mathcal{N}(0, \sigma_1^2), \tag{10}$$

$$\lambda_t^{sa} = \lambda_{t-1}^{sa} + d_t, \quad d_t \sim \mathcal{N}(0, \sigma_2^2). \tag{11}$$

AR1 process for the phase increments of the annual and semi-annual cycles:

$$\omega_t^a = \omega_m^a + \rho_1(\omega_{t-1}^a - \omega_m^a) + g_t, \quad g_t \sim \mathcal{N}(0, \sigma_3^2), \tag{12}$$

$$\omega_t^{sa} = \omega_m^{sa} + \rho_2(\omega_{t-1}^{sa} - \omega_m^{sa}) + s_t, \quad s_t \sim \mathcal{N}(0, \sigma_4^2). \tag{13}$$

Phase of the annual and semi-annual cycles:

$$\phi_t^a = \phi_{t-1}^a + \omega_t^a, \tag{14}$$

$$\phi_t^{sa} = \phi_{t-1}^{sa} + \omega_t^{sa}. \tag{15}$$

Low-frequency component:

$$b_t = b_{t-1} + v_t, \quad v_t \sim \mathcal{N}(0, \sigma_5^2), \tag{16}$$

where $\omega_m^a$ and $\omega_m^{sa}$ represent the mean frequency of the annual and semi-annual cycles, respectively, and hence their value is set equal to $2\pi/12$ and $2\pi/6$ (for monthly data). $\mathbf{x}_t = (\lambda_t^a, \lambda_t^{sa}, \omega_t^a, \omega_t^{sa}, \phi_t^a, \phi_t^{sa}, b_t)$ is the latent state at time $t$, whereas $\theta = (\rho_1, \rho_2, \sigma_0^2, \sigma_1^2, \sigma_2^2, \sigma_3^2, \sigma_4^2, \sigma_5^2)$ are the unknown parameters of the model. Equations (9–16) form our state-space model.

Bayesian inference in state-space models relies on evaluation of the joint posterior density $p(\theta, \mathbf{x}_{1:T}|y_{1:T})$, which for our non-linear model does not admit a closed-form expression. To perform inference in our model, we use a recently introduced class of algorithms named particle Markov chain Monte Carlo (MCMC) samplers[51], which enables us to sample efficiently from $p(\theta, \mathbf{x}_{1:T}|y_{1:T})$ in an MCMC. In particular, we use a state-of-the-art particle MCMC sampler referred to as particle Gibbs with ancestor sampling[52] (PGAS), which has been shown to provide rapid mixing of the Markov kernel even when using few particles in the underlying particle filter.

One special feature of our state-space model is that the state transition kernel is degenerate in the sense that the process noise associated with either $\phi_t^a$ or $\phi_t^{sa}$ is exactly zero, which renders PGAS inapplicable in its standard form. To address this issue and enable inference in our degenerate model, we use a modification of PGAS-denoted particle rejuvenation[53]. With this modification, the algorithm to sample from $p(\theta, \mathbf{x}_{1:T}|y_{1:T})$ consists of sampling iteratively from $p(\theta|\mathbf{x}_{1:T}, y_{1:T})$ and $p_\theta(\mathbf{x}_{1:T}|y_{1:T})$ as follows:

Step 1: set $\theta(0)$ and $\mathbf{x}_{1:T}(0)$ arbitrarily
Step 2: for iteration $i \geq 1$ do

 a. Draw $\theta(i) \sim p(\theta|\mathbf{x}_{1:T}(i-1), y_{1:T})$, and
 b. Sample $\mathbf{x}_{1:T}(i)$ from the PGAS Markov kernel (with particle rejuvenation) targeting $p_{\theta(i)}(\mathbf{x}_{1:T}|y_{1:T})$ conditional on $\mathbf{x}_{1:T}(i-1)$

Step 2a requires that we ascribe prior distributions to all the static parameters. For the variance parameters $(\sigma_i^2)_{i=0:5}$, we use a non-informative inverse gamma prior, $\mathcal{IG}(0.01, 0.01)$, while the AR1 coefficients ($\rho_1, \rho_2$) are assigned a uniform prior $\mathcal{U}(0, 1)$. The inverse gamma distribution as a prior for variance parameters is a standard choice in Gaussian models because it gives a closed-form expression for the posterior by virtue of its conjugate form. Setting its two hyperparameters (shape and scale) to a small number (e.g., 0.01) defines a non-informative (or weakly informative) prior that has little effect on the posterior, and thus on our inference. Therefore, it is crucial to note that all the static parameters along with

the latent state are inferred from the observations by PGAS without any manual tweaking.

For the particle filter, we use a bootstrap implementation with the number of particles set equal to the length of the time series (i.e., $T$) (note that the algorithm does not rely on asymptotics in the number of particles to be correct, however a higher number of particles improves the mixing speed of the algorithm). The number of iterations is set to 20,000 with a burn-in period of 2000. All credible intervals shown in this paper represent the highest posterior density interval (i.e., the interval with the smallest width among all the credible intervals for a specified significance level) and are computed using the Chen-Shao algorithm[54]. As an illustration, estimates of the state variables for the Key West tide gauge derived using our method along with the trace plots for the eight static parameters of the model are shown in Supplementary Fig. 5.

To demonstrate the high skill of our method, we have performed a numerical experiment on synthetic data. In particular, we have generated a synthetic time series containing a predetermined time-varying annual cycle, a low-frequency component, and realistic noise. The prescribed annual amplitude has variations similar to those observed in tide-gauge records. We have then applied our method to infer the annual amplitude from the synthetic time series and have compared our estimates with those obtained using a harmonic fit to 5-year running windows. Estimates are computed for 100 different realizations of the noise. The results are shown in Supplementary Fig. 6. The range of individual estimates based on the state-space model fully encompasses the true amplitude at all time steps. Furthermore, nearly all estimates fall within the 95% credible interval computed by PGAS, indicating that our method provides realistic uncertainty estimates. In addition, the mean of the 100 individual estimates matches the true amplitude almost exactly, meaning that our method gives unbiased estimates of the amplitude. In contrast, the range of individual estimates computed using the method of running windows do not entirely contain the true amplitude and the mean consistently underestimates the true amplitude, indicating that this method is a biased estimator of the amplitude. Note also that the method does not provide estimates within half the window size from the edges of the time series.

We have also performed a simple residual check to assess if the assumption of white-noise innovations holds. In particular, we have computed the residuals in Eqs. (9–16) based on the 18,000 samples from the posterior distribution. This results in 18,000 time series of residuals for each equation and tide-gauge station. Then, we have computed the mean AR1 coefficient over the 18,000 samples. Values of the mean AR1 coefficient fall within the range $(-0.03, 0.15)$ for all equations and tide-gauge stations, which gives us confidence in the correctness of the model.

Finally, It is worth mentioning that we have tested a slightly modified version of the model that included an AR1 term in addition to the four terms of Eq. (9), but found that such model yielded almost identical results to the model without the AR1 process. Furthermore, in most cases the MCMC chain exhibits better mixing in the simpler model. This indicates that the benefit of adding an AR1 process does not outweigh the costs of increased complexity, and thus we opted for the simpler model.

**State-space model for the UMO variance**. To estimate the instantaneous variance of the UMO transport time series, we use a stochastic variance model, which is given by the following state-space model:

$$m_t = m_{t-1} + j_t, \quad j_t \sim \mathcal{N}\big(0, \sigma_{\mathrm{m}}^2\big), \tag{17}$$

$$n_t = n_{t-1} + p_t, \quad p_t \sim \mathcal{N}\big(0, \sigma_{\mathrm{n}}^2\big), \tag{18}$$

$$u_t = \rho_1 u_{t-1} + \rho_2 u_{t-2} + h_t, \quad h_t \sim \mathcal{N}(0, \kappa), \tag{19}$$

$$y_t = \exp(n_t/2)u_t + m_t + k_t, \quad k_t \sim \mathcal{N}\big(0, \sigma_{\mathrm{y}}^2\big), \tag{20}$$

where now $y_{1:T} = (y_1,\dots,y_T)$ denote the UMO transport time series, $m_t$ represent low-frequency variations in the UMO, $u_t$ is a second-order autoregressive (AR2) process used to model the high-frequency (intra- to inter-annual) variations in the UMO, and $\exp(n_t)$ is the instantaneous variance of the AR2 process (the quantity that we use as a measure of the intensity of the UMO variations). The unknown parameters of this model are $\boldsymbol{\theta} = \big(\rho_1, \rho_2, \sigma_{\mathrm{m}}^2, \sigma_{\mathrm{n}}^2, \sigma_{\mathrm{y}}^2\big)$, and $\kappa = \frac{1+\rho_2}{1-\rho_2}\big[\big(1-\rho_2\big)^2 - \rho_1^2\big]$ to ensure that $u_t$ has variance equal to one. For this model, the latent state at time $t$ is $\mathbf{x}_t = (m_t, n_t, u_t)$. Inference in this stochastic model is performed by PGAS, in the same way as for the seasonal state-space model described in the previous section. To do this, we assign a non-informative inverse gamma prior, $\mathcal{I}(0.01, 0.01)$, to the variance parameters, and uniform prior to the coefficients of the AR2 process to ensure that stationarity conditions $(\rho_1 + \rho_2 < 1, \rho_2 - \rho_1 < 1,$ and $|\rho_2| < 1)$ are satisfied.

**Definition of the modulator**. Amplitude modulation is mathematically expressed as

$$y_{\mathrm{m}}(t) = [1 + g(t)] \underbrace{A \cos(2\pi f t)}_{y(t)}, \tag{21}$$

where $g(t)$ is a modulation signal and $y(t)$ is a periodic signal of frequency $f$ and constant amplitude $A$ (often termed the carrier in radio communications and related disciplines).

For the sake of simplicity, let us assume that $g(t)$ is a sinusoid of frequency $f_{\mathrm{m}}$ (typically $f_{\mathrm{m}} \ll f$) and constant amplitude $B < 1$. Equation (21) can then be rewritten as

$$y_{\mathrm{m}}(t) = [1 + B \cos(2\pi f_{\mathrm{m}} t + \varphi)] A \cos(2\pi f t), \tag{22}$$

which, by using the identity $\cos a \cos b = 1/2[\cos(a+b) + \cos(a-b)]$, can be rearranged as

$$y_{\mathrm{m}}(t) = y(t) + \underbrace{\frac{AB}{2}[\cos(2\pi(f+f_{\mathrm{m}})t + \varphi) + \cos(2\pi(f-f_{\mathrm{m}})t - \varphi)]}_{\text{modulator}}. \tag{23}$$

Hence, the amplitude-modulated signal, $y_{\mathrm{m}}(t)$, can be viewed as the sum of the unmodulated signal (or carrier), $y(t)$, and two sinusoids with frequencies (referred to as the upper and lower sidebands) equal to the sum and difference frequencies of the carrier and modulation signals. The sum of the two sidebands is what we refer to as the modulator. Note that in the context of this paper, $y(t)$ represents the mean SLAC because it is what we obtain if we fit an annual harmonic to the modulated SLAC.

In practice, $g(t)$ will take a more complicated form than a sinusoid. However, by Fourier analysis, any general function can be written in terms of sinusoids, which implies that $y_{\mathrm{m}}(t)$ can always be put in the form of Eq. (23), with two sidebands for each frequency component of $g(t)$.

The modulator is not a direct output of our state-space model, however it can be easily computed as:

$$m(t) = \mathrm{Res}[a^{\mathrm{a}}(t) \cos \phi^{\mathrm{a}}(t)], \tag{24}$$

where $a^{\mathrm{a}}(t)$ and $\phi^{\mathrm{a}}(t)$ are the estimates of the annual amplitude and phase provided by the state-space model, and Res is an operator denoting residual after subtraction of the mean annual cycle. To emphasize the timescales of interest, we apply a Butterworth band-pass filter (lower and higher cutoff frequencies: 1/16 and 1/8 months$^{-1}$) to the output of Eq. (24). Note that, from Eq. (23), such band-pass filter in the modulator domain is equivalent to a 4-year low-pass filter in the amplitude domain.

As an illustration, we show the power spectral density of the SLAC modulator for the Key West and St. Petersburg tide gauges (Supplementary Fig. 3). Both tide gauges display two dominant sidebands at ~10.5 and ~13.8 months, which from Eq. (23) implies that the amplitude of the SLAC has most of its energy at periods of ~7 years. The two additional sidebands at nearly 12 months represent lower-frequency (>30 years) variability of the annual amplitude. Similar spectral peaks are observed at all tide gauges.

**Statistical significance of correlations**. The significance of cross-correlations is quantified by using the non-parametric random-phase test described by ref. [55], which accounts for serial correlation in the time series. Here we use 10,000 random-phase simulations.

**Code availability**. C++ code for the state-space models is available from the corresponding author upon request.

**Data availability**. All data sets analyzed during this study are publicly available from the links provided in the Methods section. The data from the OCCAM model are available to anyone from the corresponding author upon request.

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

## Acknowledgements

We acknowledge the PSMSL for the tide-gauge data, the OCCAM, NEMO, and SODA projects for the ocean model data, the RAPID-WATCH MOC for the UMO data, the WHOI OAFlux project for the heat flux data, CMEMS for the altimetry data, and NOAA AOML for the Florida current transport time series. E.F.-W. was supported by a Leverhulme Trust Research Fellowship. Plotting was done in Python using the Matplotlib and Basemap libraries. This work has been partially supported by the Natural Environment Research Council (NERC) National Capability funding. F.L. was supported by the Swedish Research Council (ref no: 2016-04278) and the Swedish Foundation for Strategic Research (ref no: ICA16-0015). We thank Chris W. Hughes for helpful discussions.

## Author contributions

This study was conceived by F.M.C. with contributions from the other co-authors. F.M.C. designed and implemented the state-space models, coded the reduced gravity model, performed the data analysis, and produced the figures. T.W. provided the time series of the annual amplitude based on the windowing method. F.L. aided with the implementation of PGAS and the development of the state-space models. J.W. provided the Matlab code to read the OCCAM data. E.F.-W. assisted with the calculation of the UMO.

F.M.C. wrote the paper with input from the other co-authors. All authors discussed the results and implications and commented on the paper.

## Additional information

**Competing interests:** The authors declare no competing interests.

