## [Peer Review File · Nature Communications]

Reviewers' comments:

Reviewer #1 (Remarks to the Author):

A review of "Unprecedented coherent fluctuations in the sea-level annual cycle along the United States Gulf and southeast coasts" by Calafat, Wahl, Lindsten, Williams, and Frajka-Williams.

In this paper, the authors use observational and modeling approaches to study interannual and decadal variations in the amplitude and phase of the annual cycle in mean sea level along the United States southeast coast. Their major conclusions include:

- (1.) The mean sea level annual amplitude varies coherently along the coast, from Cape Hatteras to the Yucatan.
- (2.) Annual sea level amplitude fluctuations just east of the Bahamas are related to westward propagating steric Rossby wave signals from the ocean interior.
- (3.) Sea level annual amplitudes along the South Atlantic Bight vary (roughly) in phase with variations in the Gulf Stream transport's variance, which both appear to have a common forcing mechanism.

I very much enjoyed reading this well-written manuscript. The authors address a number of topics highly relevant to oceanography and climate science, such as the nature of mean sea level variation along the coast, and its relation to the general circulation. They make use of a variety of approaches; the Bayesian inferential framework is particularly powerful, novel, and attractive. The results will contribute substantially to sea level science, moving the field forward and spurring additional research; the results will also be of broad interest to scientists in other fields as well as the general public. I believe that this paper should eventually be published in Nature Communications. However, I have a number of concerns, detailed below, that should be addressed prior to publication. I recommend returning the manuscript to the authors for minor revisions.

Specific comments—

Lines 1-2. Title: I'm not a fan of the emphasis, here and elsewhere, on some sense of "unprecedentedness" in sea level behavior. To my mind, this detail is tangential to what I think are the much more important and robust results relating to the spatial coherence of the annual amplitude fluctuations, their relation to open-ocean Rossby waves, and their correlation to overturning transport. I suggest the authors to consider the alternative title (which I believe is more reflective of the contents of the paper): Mechanisms of coherent fluctuations in the sea-level annual cycle along the United States Gulf and southeast coasts and their relation to the overturning circulation

Lines 25-26. A key result is that changes in sea level annual amplitude are correlated with time series of instantaneous overturning transport variance. The authors argue that these sea level and overturning circulation behaviors both have a common origin related to Rossby waves propagating westward from the interior. This important finding directly contradicts previous studies (e.g., by Ezer and colleagues) that claim (I believe erroneously) that changes in the Gulf Stream in some sense "cause" coastal sea level changes. I think the authors should emphasize this aspect more directly.

Line 50. "Knowing how to model and predict these seasonal changes ..." One of the main motivations of this work, stated here and elsewhere, is the desire to provide more accurate sea level predictions. Do results here have any tangible, concrete implications for the prediction problem? If so, it would be beneficial to draw out those implications more clearly.

Line 78. What is meant by "customary"?

Lines 135-136. Suggest deleting the phrase, “but it is even more apparent for changes in the SLAC,” both because this point is tangential, and also because it is not apparent from the results shown.

Lines 139-140ff. “... and unprecedentedly large values since the 1990s ...” I take issue, here and elsewhere, with this focus on some sense of “unprecedented” sea level behavior. First, I don’t believe that results shown here (Figure 1a) demonstrate that there have been unprecedentedly large values “since” the 1990s. Rather, the Figure shows that there was one large, unprecedented event, which was discussed earlier by the second author and his colleagues, around 2009. Since then, instantaneous annual amplitudes have returned to “normal,” showing that the 2009 behavior was a transient event, and not indicative of a secular change. Second, as I mentioned earlier, I think that this preoccupation with particular events is somewhat distracting, given that much more interesting process results in the paper.

Lines 148-150. “... Part of a succession of decadal fluctuations with increasingly high peaks ... ” Similar to my comment above, I believe that this wording and emphasis is both distracting and untrue. Particularly, Figure 2 shows that there were an exceptional number of exceedances for one limited, recent time interval. To my eye, the histograms do not indicate that there is any sort of secular behavior, such that number of exceedances has been growing over time. Rather, the histograms look like one extreme event superimposed on noise.

Line 170ff. Regarding Figure 4 and discussion. The interesting behavior in terms of annual sea level amplitudes from tide gauges manifests most clearly on decadal time scales. However, the data and model products used to produce Figure 4 are comparatively short (~2 decades or less). I’m wondering if similar results are found if, say, a longer model product is considered—maybe something like the SODA class of models? In other words, is Figure 4 frequency-band dependent or dependent of the record length of the product used to make the plots?

Lines 183-184. “... At speeds of a few m/s, have an offshore length scale of about 50 km...” Suggest to clarify that the authors are alluding to first baroclinic mode waves with some assumed depth/stratification structure.

Lines 193-195. “The changes in the annual amplitude ... with those from tide-gauge records.” I think it would be helpful to the reader to show a plot of this result, e.g., time series of the model and tide gauge annual amplitudes.

Lines 205-207. “The thermocline movements associated with baroclinic Rossby waves are observed as the steric component of sea level...” Baroclinic Rossby waves also manifest as anomalies in ocean bottom pressure.

Lines 210-211. I’m curious—do instances of highest correlation relate more to a particular isobath or to regions of strongest bathymetric gradient?

Line 222ff. For $\int_{\eta_s^{\text{upper}}}^{\text{bottom}} \rho \, dz$, it would be helpful to give the depth range over which the density integral is taken; that is, what depth range corresponds to the “upper thermocline” in the calculations?

Lines 235-242. The language of carrier signal and wave superposition is an interesting take. It would be helpful for the reader to include some discussion of these concepts in the methods section. Particularly, it would be informative to describe how the “steric modulator” (Figure 7) is calculated.

Lines 247-252ff. A major finding, first introduced here, is that signals along the western boundary

originate from Rossby wave signals over the interior. A very important issue, which the authors don't even touch on, is the origin of the Rossby waves. What generates them? Is it local or remote wind curl variations, or perhaps some other generating mechanism? I don't think it's necessary that the authors definitively identify the origin or mechanism, but the issue should at least be raised, and maybe some hypotheses given.

Lines 253-255. This is an extremely important sentence, which posits a causal link between the open ocean Rossby waves and coherent coastal sea level behavior. However, the sentence is highly speculative, in the sense that there are several causal "holes" that need to be filled or at least acknowledged. At this point (Figure 7), the authors haven't "reached the coast" yet. Rather, they have shown that open ocean signals are apparent east of the Bahamas. But, how does that signal there get communicated to the U.S. coast? How does the Rossby wave signal, for example, interact with the western boundary current? How does it get communicated across the Florida Straits/Gulf Stream? Is 26.5N the only relevant latitude? Or is this process occurring at other latitudes? More fleshing out is needed here.

Lines 296-299. The authors offer an intriguing hypothesis regarding how the signal gets communicated to more northern latitudes (which is opposite to the sense of coastal waves). However, it's fairly speculative. I'm wondering if the 8-15 month lag is consistent with the advection and wave time scales implied by the mechanisms being proposed.

Line 315. Suggest changing "demonstrated" to "suggested".

Line 325. Suggest changing "amplitudes" to "amplitude" (singular). Again, the only real "unprecedented" behavior I see in the results is the event ~2009 noted earlier by Wahl et al., which doesn't appear to be any secular behavior, but rather is a transient "blip."

Lines 339-340. What is the time period covered by the OAFflux product?

Line 358. Change "fee" to "free". Also, a model need not omit data assimilation to be freely running. That is, there are numerous physically consistent, freely running, data assimilating model products (e.g., the ECCO state estimates).

Line 394 and Equation 5. Please clarify that PE and PW are ocean bottom pressures.

Lines 412ff. It is confusing and misleading for the authors to say that, "two equations are required." Many more than two equations are given in the development of the Bayesian models. Do the authors mean two "levels" (data and process)? If so, since the authors are making inference on the uncertain model parameters, really it should be three levels (data, process, parameter).

Line 425 and Equation 7. I find it odd and non-standard that the authors represent the "physics" in the measurement model (equation 7). Usually, the measurement model represents a mapping between the hidden process (sea level) and the noisy data (tide gauge record), and the "physics" are prescribed in the process level. Can the authors explain their "unorthodox" modeling choice here?

Equations 7-14. In most of these equations, a white noise innovation sequence is assumed. Is this a good assumption? Is it warranted given the data? For interpreting the Bayesian model results, it would be very informative to provide – in addition to the nice "checks and balances" already provided by the authors (e.g., Figures S2, S3) – a formal residual analysis, that will demonstrate (or not) if the residuals obey the assumed white noise behavior.

Line 490. What is the justification for the priors on all the inverse gamma distributions? For the unacquainted reader, it would be helpful to explain what the parameters are that control the inverse gamma distribution, and to discuss why the particular numerical values (0.01, 0.01) are chosen. Also, units?

Lines 504-506. I very much appreciate Figure S3, which gives a tangible sense of the model performance. However, the phrase "with great confidence" is too qualitative for my likings. One of the advantages of the fully Bayesian approach used here is that it provides a very thorough and rigorous quantification of uncertainty. I'm wondering how meaningful the uncertainty estimates furnished by the posterior solution are. For example, are 95% of the true values captured by the 95% credible intervals? This is something that could be easily quantified with the output from the synthetic data experiments. Having some sense of the meaningfulness of the estimated errors would be helpful for interpreting the results.

Line 699 and Figure 1a and general comments on the Bayesian model framework. It strikes me, looking at Figure 1a and glancing over the model equations, that the authors have selected to model each tide gauge time series in isolation. This is surprising, given that there is very strong spatial coherence. That is, each tide gauge provides informative constraints on the behavior at other tide gauge sites. What is the justification for, in the model design, choosing to model each location in isolation, rather than representing sea level as a process with spatial and temporal covariance, which would allow the many gaps in the data (cf. Figure 1a) to be filled, thus providing a complete estimation at all sites going back to ~1900?

Figure 2. I'm confused. Why does the 68% credible interval span negative values at site 12?

Figure 4. Black dots indicating statistical significance are difficult to see visually.

Figure 5. How sensitive is this correlation spatial pattern to the choice of depth range bottom (here 500 m). Are similar results found if a different, shallower depth (e.g., 200 m) is used?

Figure 8. Related to my earlier comment about the authors having chosen to model each tide gauge in isolation, rather than as a spatiotemporal process, I wonder how robust the uncertainty estimates are on the across-site average time series in Figure 8. Specifically, given the model design, the posterior draws of the sea level solution at one site will be uncorrelated with the posterior draws of sea level at another site. In reality, the draws should be correlated, given the strong spatial coherence, e.g., in Figure 1a.

Reviewer #3 (Remarks to the Author):

Review of "Unprecedented coherent fluctuations in the sea-level annual cycle along the United States Gulf and southeast coasts" by Calafat et al.

This manuscript is not suitable for publication in its current form.

Major comments:

I find the title misleading. What is reported in the manuscript is decadal oscillations of coherent fluctuations in SLAC along the US coast.

The correlation map from OCCAM (Fig. 4) shows significant point-wise correlations not confined to the

south of Cape Hatteras (as in satellite altimetry), but extend much further to the north.

The authors dismiss the possibility of boundary wave generation by wind forcing along the coast without showing any supporting evidence.

No significant correlations are found between the steric height in the open ocean and that averaged along the coast (Fig. 5). How can the authors then claim that the coherent SLAC along the coast is generated by westward-propagating Rossby waves impinging on the western boundary?

The selection of the very small yellow box in Figure 5 for investigating the role of Rossby waves is highly subjective and not satisfactory.

The Hovmoller diagram in Fig. 7 only shows anomalies propagating westward at 26.5N. It does not say anything about the connection between sea level variability along the US coast and sea level variability at this latitude.

Using advection of anomalies by the Gulf Stream to explain SLAC coherence along the coast is not convincing. It is purely speculation.

Reviewer #4 (Remarks to the Author):

Review on manuscript (#NCOMMS-17-24109) "Unprecedented coherent fluctuations in the sea-level annual cycle along the United States Gulf and southeast coasts" by Calafat et al.

This manuscript examined the temporal variations in the sea-level annual cycle amplitude along the US Gulf and southeast coasts using observations and ocean model simulation, and found unprecedentedly large decadal fluctuations since 1990. It's suggested that the incident Rossby waves from the basin interior is the main mechanism. The topic is an interesting one to the sea-level research community, and may have implication for short-term coastal planning. The manuscript is also written well. However, I have some major concerns/questions about the main points of this paper. Detailed comments are as follows.

1. This is a general comment about definition of annual cycle and its temporal variations. What does annual cycle mean? Over which period can an annual cycle be robustly defined? Obviously, the derivation of annual cycle, e.g., through the traditional least-squares fitting, should have some sensitivity to the analysis period (starting/ending time, length, etc). Different periods may give different annual cycles. So, in many ocean/climate related studies, it's critical to define a "base-period", usually long enough (>20 years), so that an annual cycle (or closely related monthly climatology) can be derived, and any deviations from it are called anomalies. For example, the popular NOAA Optimum Interpolation Sea Surface Temperature (OISST, <https://www.ncdc.noaa.gov/oisst>) product define SST anomaly relative to a climatology over a 30 years base period (1971-2000). Quite strong interannual and decadal variations of sea level annual cycle amplitude as claimed by this study (e.g., shown in Fig. 3 and Supplementary Fig. 2a), in fact may be just interannual and decadal variations of sea level. For ENSO and decadal variability research community, they would have a totally different interpretation of your Fig. 6a by assuming a repeating annual cycle over the whole analysis period and discussing deviations from this repeating annual cycle as interannual & decadal variations of sea level. Can you repeat your analysis following above practice? How would it affect your interpretation?

2. The section introducing state-space model for the annual cycle in the Methods is too technical for

most readers of Nature Communications. The example shown in the Supplementary Figure 2 is helpful, but is not enough. You need to simplify it, explain the pros/cons of this method, and its comparison with methods used by others (e.g., Wahl et al., ref. 16).

3. Thompson and Mitchum (2014) identified coherent sea level variability in the Gulf coast and US east coast, and explained the coherence is due to basin-scale zonal redistributions of volume (see their section 5). Should this mechanism also work to explain the coherence identified in this study?

Thompson, P. R., and G. T. Mitchum (2014), Coherent sea level variability on the North Atlantic western boundary, *J. Geophys. Res. Oceans*, 119, 5676–5689, doi:10.1002/2014JC009999.

Minor comments

4. Line 358, fee-running should be free-running

Response to Reviewers

We thank the reviewers for their valuable and insightful comments. In the following, we explain how the manuscript has been changed and provide a point-by-point response to each of the reviewers' comments. For clarity, our responses are in blue whereas any text from the reviews is denoted by black italic font.

We have substantially revised the manuscript by performing additional analysis, including new datasets, and clarifying our arguments. We will first describe the main changes made to the manuscript and will address the reviewers' comments involving the mechanism underlying the changes in the sea-level annual cycle (SLAC).

1. In the original manuscript, we showed that the mean SLAC and its modulation are associated with steric contributions from different ocean layers (above and below the seasonal thermocline). This is a general result in that it holds regardless of the underlying mechanisms. This result has major implications for the understanding of the SLAC modulation, yet it went somehow unnoticed. Furthermore, from the reviews, it seems that there was some confusion about what such result meant or how we exploited it. Given its importance, we should have more strongly emphasized this point. A new paragraph discussing this result as well as a new subsection in Methods explaining the concept of modulator have been added to the revised manuscript. In addition, we have added a schematic illustration (new Fig. 8) explaining the proposed mechanism. The new paragraph reads:

“Two implications can be drawn. First, all the information on the modulation of the SLAC resides in the ocean layers below the seasonal thermocline. Second, the SLAC can be simply described as the sum of the unmodulated cycle and a term representing steric changes below the seasonal thermocline (hereafter referred to as the modulator):

$$\text{SLAC} = \underbrace{\text{SLAC}_{\text{mean}}}_{\substack{\text{Steric above} \\ \text{seasonal thermocline}}} + \underbrace{\text{SLAC}_{\text{modulator}}}_{\substack{\text{Steric below} \\ \text{seasonal thermocline}}}$$

These two implications affect how we understand the SLAC modulation and provide the basis for the subsequent analysis. In this regard, note the following. In the time domain, amplitude modulation typically involves multiplication of a low-frequency modulating signal and a high-frequency sine wave (the latter is often termed the ‘carrier’ in radio communications). However, from properties of the Fourier transform, multiplication in the time domain corresponds to convolution in the frequency domain. Therefore, in the frequency domain, amplitude modulation appears as sums and differences of the frequencies of the two input signals instead of as products. This implies that any modulated signal can always be mathematically described as the sum of the carrier and a superposition of sinusoids with frequencies slightly above and below the carrier frequency (see Methods for proof). This alternative interpretation is exactly analogous to the steric representation of the SLAC modulation. It turns out that the ocean, along its vertical dimension, behaves similarly to a Fourier transform in that it separates the frequency components of the SLAC into different ocean layers. This result will greatly facilitate our analysis.”

2. Two of the reviewers raised questions about the proposed mechanism of SLAC modulation. In particular, they asked for stronger evidence to support our conclusions. We are not sure whether the reviewers were asking for a detailed explanation of how Rossby waves interact with sloping topography when they reach the continental slope. The ocean-to-shelf transmission problem is a complex issue that has been studied before (e.g., Huthance, 2004) and is currently the focus of intensive research, but which remains a difficult scientific challenge. Hence, though we recognise its importance, a detailed analysis is beyond the scope of this paper. Instead, we propose an explanation for the spatio-temporal pattern of the SLAC that to us seems the most plausible given the observations, numerical models, and the general sea-level theory. In the original manuscript, we

showed that the SLAC modulator along the Gulf and Southeast coasts is correlated with the modulator at the Bahamas east coast (note how similar the two time series are in the left panel of Figure 7), and that both are consistent with density anomalies propagating westward as Rossby waves (see Hovmöller diagram in Figure 7). Therefore, we concluded that the modulation of the SLAC is controlled by incident Rossby waves. We believe that our original analysis provided solid evidence for the proposed mechanism. Nevertheless, after considering this issue at length, we recognize that the reviewers' concerns are fair and have responded by performing additional analysis and clarifying our arguments as described below.

- We have included data from the NEMO (1/4°) global ocean model (1968-2012) in our analysis.
- We have produced lagged correlation maps showing the correlation of the steric contribution from 200 to 1000 m depth at the continental slope east of the Bahamas with the steric contribution at each grid point in both OCCAM and NEMO. For grid points in shallow areas (<200 m) the correlation is computed with the SLAC modulator instead of the steric. The pattern of evolution (new Figure 6) from both models shows a region of significant correlation several hundred kilometers off the coast of the Bahamas at lags of ~3 months, indicating a lagged relationship between this region and the western boundary. As the lag decreases, the region of correlation propagates westward until it reaches the coast at lag zero and then the entire shelf and coastal zone become significantly correlated, both in the Gulf of Mexico and along the Southeast coast.
- To assess how the density anomalies are modified by the continental slope as they approach the coast, we have computed the standard deviation of the SLAC modulator as a function of distance from the Bahamas coast along with the correlation between the modulator at the coast and that offshore (new Fig. 7b). We find that, while there is a gradual decrease in the magnitude of the modulator with proximity to the boundary, the phase coherence remains significant through the continental slope. The reduction in dynamic height variability towards the western boundary has been reported before and is explained by frictional energy dissipation and the export of energy through boundary waves. The latter is precisely the mechanism that we invoke to explain the coherence of the amplitude over large distances along the coast. Just as longshore wind can excite boundary waves by depressing or lifting the thermocline near the boundary, so can Rossby waves incident on a boundary.
- Reviewer #1 wondered if the correlation maps in the original Figure 4 are “*frequency-band dependent or dependent of the record length of the product used to make the plots*”. This is a valid point. We have produced additional correlation maps (new Figs. 4c,d) based on NEMO (1968-2012) and the SODA reanalysis (1900-2010). The spatial patterns of correlation derived from SODA and NEMO are very similar to those from both altimetry and OCCAM. This confirms that the pattern is not dependent on the period or its length, which gives us additional confidence in our results.

3. Reviewers #1 and #3 feel that the use of the word ‘unprecedented’ in the title and elsewhere is unjustified. We did not intend to give the impression of a long-term trend in the amplitude of the SLAC in recent decades, and we made this clear in the original manuscript by writing (lines 158-160): “*the annual amplitude fell back to average values after 2009 as part of a large decadal oscillation (Fig. 3), limiting support for the existence of a long-term trend*”. Instead, what we reported is a large decadal fluctuation in 2008-2009 that seemed unprecedented. To quantify how unique this fluctuation is we have analysed the histograms shown in Fig. 2 by comparing the values in 2008-2009 with those before 1990, taking into account their credible intervals. We have concluded that the event of 2008-2009 is *likely* (probability $\geq 68\%$) unprecedented in the tide-gauge record as indicated by the non-overlapping credible intervals. To support this point, two error bars have been added to the histograms of Fig 2, representing the 68% credible intervals associated with

the maximum value in 2008-2009 and in the period before 1990. This result is included in the abstract, however we agree that the word unprecedented in the title may be misleading, and hence we have removed it. The title now reads:

“Mechanisms of coherent fluctuations in the sea-level annual cycle along the United States Gulf and southeast coasts”

Reviewer #1

A review of “Unprecedented coherent fluctuations in the sea-level annual cycle along the United States Gulf and southeast coasts” by Calafat, Wahl, Lindsten, Williams, and Frajka-Williams.

In this paper, the authors use observational and modeling approaches to study interannual and decadal variations in the amplitude and phase of the annual cycle in mean sea level along the United States southeast coast. Their major conclusions include:

(1.) The mean sea level annual amplitude varies coherently along the coast, from Cape Hatteras to the Yucatan.

(2.) Annual sea level amplitude fluctuations just east of the Bahamas are related to westward propagating steric Rossby wave signals from the ocean interior.

(3.) Sea level annual amplitudes along the South Atlantic Bight vary (roughly) in phase with variations in the Gulf Stream transport’s variance, which both appear to have a common forcing mechanism.

I very much enjoyed reading this well-written manuscript. The authors address a number of topics highly relevant to oceanography and climate science, such as the nature of mean sea level variation along the coast, and its relation to the general circulation. They make use of a variety of approaches; the Bayesian inferential framework is particularly powerful, novel, and attractive. The results will contribute substantially to sea level science, moving the field forward and spurring additional research; the results will also be of broad interest to scientists in other fields as well as the general public. I believe that this paper should eventually be published in Nature Communications. However, I have a number of concerns, detailed below, that should be addressed prior to publication. I recommend returning the manuscript to the authors for minor revisions.

We thank the reviewer for his/her constructive and detailed review of our manuscript.

Specific comments—

Lines 1-2. Title: I’m not a fan of the emphasis, here and elsewhere, on some sense of “unprecedentedness” in sea level behavior. To my mind, this detail is tangential to what I think are the much more important and robust results relating to the spatial coherence of the annual amplitude fluctuations, their relation to open-ocean Rossby waves, and their correlation to overturning transport. I suggest the authors to consider the alternative title (which I believe is more reflective of the contents of the paper): Mechanisms of coherent fluctuations in the sea-level annual cycle along the United States Gulf and southeast coasts and their relation to the overturning circulation

Please see the discussion at the beginning of this letter (Point 3). As discussed, the title has been revised and now reads:

“Mechanisms of coherent fluctuations in the sea-level annual cycle along the United States Gulf and southeast coasts”

We have decided not to include a reference to the overturning circulation in the title for two reasons.

First, because we are not keen on long titles and the title is already quite long. Second, because the link between the amplitude of the SLAC and the upper mid-ocean transport does not imply that one drives the other, but rather that both quantities are affected by the same density anomalies at the western boundary of the Atlantic.

Lines 25-26. A key result is that changes in sea level annual amplitude are correlated with time series of instantaneous overturning transport variance. The authors argue that these sea level and overturning circulation behaviors both have a common origin related to Rossby waves propagating westward from the interior. This important finding directly contradicts previous studies (e.g., by Ezer and colleagues) that claim (I believe erroneously) that changes in the Gulf Stream in some sense “cause” coastal sea level changes. I think the authors should emphasize this aspect more directly.

We believe that the result to which the reviewer is referring is already clearly emphasized in the abstract. In addition, while we share the reviewer’s intuition that the contribution from changes in the Gulf Stream to sea-level variability is relatively minor, our work is about variations in the amplitude of the SLAC and as such our results should not be extrapolated to other timescales without further analysis. In that sense, statements such as that our findings contradict those from previous studies on the role of the Gulf Stream at other timescales are not really supported by our analysis.

Line 50. “Knowing how to model and predict these seasonal changes ...” One of the main motivations of this work, stated here and elsewhere, is the desire to provide more accurate sea level predictions. Do results here have any tangible, concrete implications for the prediction problem? If so, it would be beneficial to draw out those implications more clearly.

Several of the authors of this manuscript are currently working on probabilistic approaches for improved seasonal forecasts of coastal sea levels and so we have a strong interest in the prediction problem. Based on our work so far on this subject, we envision several ways how our results may be used for improved predictions. For example, we find that most of the energy in the SLAC modulator is at periods of ~10.5 months (see new Supplementary Fig. 2). This can be used to build probabilistic models of harmonic structure for the SLAC modulator. In fact, preliminary analysis shows that such simple models already outperform more ‘naïve’ persistence models. Our results on the mechanisms tell us where to focus if we want to incorporate physical information into our probabilistic models. For instance, we have shown that all the information about the modulation resides in the layers below the seasonal thermocline and that such information originates in the ocean interior. Although, to exploit this knowledge we first need to understand how to model the reduction in variance of the density anomalies as they approach the coast (see new Fig. 7b). However, these ideas have not been tested and so we are very reluctant to make statements about concrete implications of our results that are not supported by quantitative analysis. We admit that we are rather vague when talking about prediction, but we are purposely so. If the reviewer insists that we should be more concrete on this topic then we would make an effort to elaborate on it, otherwise we opt for leaving it as is.

Line 78. What is meant by “customary”?

By customary we meant that such approach is widely used in sea-level analyses. In the new version of the manuscript we have replaced this word with “commonly used”.

Lines 135-136. Suggest deleting the phrase, “but it is even more apparent for changes in the SLAC,” both because this point is tangential, and also because it is not apparent from the results shown.

We agree. This sentence has been removed from the manuscript.

Lines 139-140ff. “... and unprecedentedly large values since the 1990s ...” I take issue, here and elsewhere, with this focus on some sense of “unprecedented” sea level behavior. First, I don’t

believe that results shown here (Figure 1a) demonstrate that there have been unprecedentedly large values “since” the 1990s. Rather, the Figure shows that there was one large, unprecedented event, which was discussed earlier by the second author and his colleagues, around 2009. Since then, instantaneous annual amplitudes have returned to “normal,” showing that the 2009 behavior was a transient event, and not indicative of a secular change. Second, as I mentioned earlier, I think that this preoccupation with particular events is somewhat distracting, given that much more interesting process results in the paper.

Please see the discussion at the beginning of this response (Point 3).

Lines 148-150. “... Part of a succession of decadal fluctuations with increasingly high peaks ... ” Similar to my comment above, I believe that this wording and emphasis is both distracting and untrue. Particularly, Figure 2 shows that there were an exceptional number of exceedances for one limited, recent time interval. To my eye, the histograms do not indicate that there is any sort of secular behavior, such that number of exceedances has been growing over time. Rather, the histograms look like one extreme event superimposed on noise.

We agree with the reviewer. Please see the discussion at the beginning of this response (Point 3).

Line 170ff. Regarding Figure 4 and discussion. The interesting behavior in terms of annual sea level amplitudes from tide gauges manifests most clearly on decadal time scales. However, the data and model products used to produce Figure 4 are comparatively short (~2 decades or less). I’m wondering if similar results are found if, say, a longer model product is considered—maybe something like the SODA class of models? In other words, is Figure 4 frequency-band dependent or dependent of the record length of the product used to make the plots?

As we describe at the beginning of this letter (Point 2), we have included additional correlation maps based on longer numerical simulations that show almost identical spatial patterns to altimetry and OCCAM.

Lines 183-184. “... At speeds of a few m/s, have an offshore length scale of about 50 km...” Suggest to clarify that the authors are alluding to first baroclinic mode waves with some assumed depth/stratification structure.

Done.

Lines 193-195. “The changes in the annual amplitude ... with those from tide-gauge records.” I think it would be helpful to the reader to show a plot of this result, e.g., time series of the model and tide gauge annual amplitudes.

Following the reviewer’s suggestion, we have added a new Figure (Supplementary Fig. 3) where we show the amplitude time series for the tide gauges, OCCAM, and the NEMO model. We also show the correlation between the modelled and observed amplitudes for all individual stations. Both models show significant correlations at most tide-gauge stations, though OCCAM performs better than NEMO.

Lines 205-207. “The thermocline movements associated with baroclinic Rossby waves are observed as the steric component of sea level...” Baroclinic Rossby waves also manifest as anomalies in ocean bottom pressure.

We agree. We did not intend to mean that Rossby waves have no imprint on bottom pressure but rather that, in the present context, their contribution to the steric height is significantly larger. Nevertheless, this sentence has been removed from the revised manuscript.

Lines 210-211. I'm curious—do instances of highest correlation relate more to a particular isobath or to regions of strongest bathymetric gradient?

The highest correlations are found along the continental slope but do not exactly refer to a particular isobaths. Therefore, we have removed the reference to the 300m isobath.

Line 222ff. For η_s^{upper} , it would be helpful to give the depth range over which the density integral is taken; that is, what depth range corresponds to the “upper thermocline” in the calculations?

The reference level that we use to compute the steric contribution from above the seasonal thermocline is about 70 m, which is, in fact, the average depth of the mixed layer in the model. We have tested other values in the range 60-120 and found no significant differences in our results. The reference depth is now indicated both in the text and in the Methods (Sea-level equations).

Lines 235-242. The language of carrier signal and wave superposition is an interesting take. It would be helpful for the reader to include some discussion of these concepts in the methods section. Particularly, it would be informative to describe how the “steric modulator” (Figure 7) is calculated.

As discussed at the beginning of this letter (Point 1), a new subsection in Methods explaining the concept of modulator and how it is calculated has been added to the revised manuscript.

Lines 247-252ff. A major finding, first introduced here, is that signals along the western boundary originate from Rossby wave signals over the interior. A very important issue, which the authors don't even touch on, is the origin of the Rossby waves. What generates them? Is it local or remote wind curl variations, or perhaps some other generating mechanism? I don't think it's necessary that the authors definitively identify the origin or mechanism, but the issue should at least be raised, and maybe some hypotheses given.

We agree that this is an important issue, though one that has been studied before. We have responded by including two new references where mechanisms of Rossby wave generation are discussed:

Cabanes, C., T. Huck, and A. C. deVerdière 2006. Contributions of wind forcing and surface heating to interannual sea level variations in the Atlantic Ocean, *J. Phys. Oceanogr.*, 36(9), 1739–1750.

Piecuch, C. G., and R. M. Ponte, 2012: Buoyancy-driven interannual sea level changes in the southeast tropical Pacific. *Geophys. Res. Lett.*, 39, L05607.

These two studies show that Rossby waves can be generated by either wind-stress curl or buoyancy forcing. Identifying which of these two forcing factors underlies the Rossby waves discussed in the present paper is interesting but is not the focus of the paper.

Lines 253-255. This is an extremely important sentence, which posits a causal link between the open ocean Rossby waves and coherent coastal sea level behavior. However, the sentence is highly speculative, in the sense that there are several causal “holes” that need to be filled or at least acknowledged. At this point (Figure 7), the authors haven't “reached the coast” yet. Rather, they have shown that open ocean signals are apparent east of the Bahamas. But, how does that signal there get communicated to the U.S. coast? How does the Rossby wave signal, for example, interact with the western boundary current? How does it get communicated across the Florida Straits/Gulf

Stream? Is 26.5N the only relevant latitude? Or is this process occurring at other latitudes? More fleshing out is needed here.

Please see our discussion at the beginning of this document (Point 2). Regarding the question as to whether this process is occurring at other latitudes, we have added the following paragraph to the revised manuscript:

“It should be noted that there is a theoretical upper limit on the frequency of Rossby waves beyond which such waves cannot exist. This limit follows from the dispersion relation and for long Rossby waves varies with latitude according to $\frac{1}{R} \sqrt{g \beta}$, where c is the baroclinic gravity-wave phase speed, R is the radius of the earth, and θ denotes latitude. The dependence on latitude imposes a constraint on where Rossby waves might act as the SLAC modulator because this possibility requires waves with nearly annual periods. In particular, spectral analysis of the modulator reveals an upper sideband of ~ 10.5 months at all tide-gauge stations (Supplementary Fig. 2). It follows then that Rossby waves with periods of ~ 10.5 months are required, but these are only possible at latitudes south of $\sim 40N$.”

Lines 296-299. The authors offer an intriguing hypothesis regarding how the signal gets communicated to more northern latitudes (which is opposite to the sense of coastal waves). However, it’s fairly speculative. I’m wondering if the 8-15 month lag is consistent with the advection and wave time scales implied by the mechanisms being proposed.

We agree with the reviewer that the effect on the Gulf Stream is a conjecture and so we expressed it in the original manuscript. However, this conjecture is based on previous published studies (Frajka-Williams et al., 2013, 2016) showing an instantaneous response of the Gulf Stream to incident density anomalies east of the Bahamas. This is consistent with the coherence of the annual amplitude found at higher latitudes, given the link between the annual amplitude and density anomalies east of the Bahamas. It is also consistent with what the new Fig. 6 shows. We have tried to explain things more clearly in the revised manuscript. The corresponding paragraph now reads (original Fig. 9 has been removed):

“This would be consistent with results from previous studies that showed a significant response of the Gulf Stream to incident density anomalies from the ocean interior. In particular, they found that, on the timescales relevant to the SLAC modulator (\sim annual), the Florida Current responds almost instantaneously to incident density anomalies just east of the Bahamas leading to a significant anti-correlation with the UMO. This response of the Florida Current could explain the coherence of the SLAC amplitude at high latitudes. In support of this premise, we find that the SLAC modulator from tide gauges along the Southeast coast (stations 11-14) is correlated (-0.34 , significant at the 95% confidence level) with band-pass filtered ($1/20 - 1/5$ months $^{-1}$) variations of the Florida Current transport.”

Line 315. Suggest changing “demonstrated” to “suggested”.

Done.

Line 325. Suggest changing “amplitudes” to “amplitude” (singular). Again, the only real “unprecedented” behavior I see in the results is the event ~ 2009 noted earlier by Wahl et al., which doesn’t appear to be any secular behavior, but rather is a transient “blip.”

We have changed “amplitudes” to “amplitude”. Regarding the concerns about the word “unprecedented”, please see our discussion at the beginning of this letter.

Lines 339-340. What is the time period covered by the OAFlux product?

OAFlux data cover the period 1983-2009. This information has been added to the revised manuscript.

Line 358. Change “fee” to “free”. Also, a model need not omit data assimilation to be freely running. That is, there are numerous physically consistent, freely running, data assimilating model products (e.g., the ECCO state estimates).

We thank the reviewer for catching this typo, which has been corrected in the revised manuscript. It seems fairly standard to refer to models that do not assimilate data as “free-running”. Nevertheless, following the reviewer’s comment we have slightly reworded this sentence, which now reads: “free-running (without data assimilation)”.

Line 394 and Equation 5. Please clarify that PE and PW are ocean bottom pressures.

Ideally they are bottom pressure, but note that in our calculation we use a vertical profile near the western boundary and thus $PW(z)$ is not bottom pressure. The transport due to the flow between the western vertical profile and the Bahamas east coast is estimated directly from the model velocities, as discussed in the manuscript.

Lines 412ff. It is confusing and misleading for the authors to say that, “two equations are required.” Many more than two equations are given in the development of the Bayesian models. Do the authors mean two “levels” (data and process)? If so, since the authors are making inference on the uncertain model parameters, really it should be three levels (data, process, parameter).

We thank the reviewer for pointing out this confusing phrase. As the reviewer has correctly guessed, what we mean is that two levels, or types, of equations are used; those that govern the dynamics of the latent-state process and those that describe how the observations relate to the state of the system. It is customary in the literature on state-space models to say that the model is described by these two (types of) equations, however the reviewer is right that the prior distribution over the parameters is also part of the model. We agree that this can be confusing, especially for those unfamiliar with the subject, and thus we have replaced “two equations” with “two types of equations” to avoid confusion.

Line 425 and Equation 7. I find it odd and non-standard that the authors represent the “physics” in the measurement model (equation 7). Usually, the measurement model represents a mapping between the hidden process (sea level) and the noisy data (tide gauge record), and the “physics” are prescribed in the process level. Can the authors explain their “unorthodox” modeling choice here?

We agree that the “physics” (i.e., the dynamical evolution) of the system should be described by the state process. However, this is in fact the case in the model that we consider; see Equations 8-14. Equation 7 simply describes how the state of the hidden process (the sea level, decomposed into amplitudes, phases, and low-frequency) relates to the measurement, but it contains no physics. The hidden variables in our state-space model are the amplitude and phase of the annual and semi-annual cycles, and the low-frequency component. The time evolution of the hidden variables is defined by the transition model (Equations 8-14), whereas the measurement model simply describes how the hidden variables are observed (or, equivalently, how the observations can be generated from the hidden variables). For example, the amplitude and phase of the annual cycle are observed as a modulated sine wave, and so Equation 7 expresses it. Hence, our model has a standard form as written.

Equations 7-14. In most of these equations, a white noise innovation sequence is assumed. Is this a good assumption? Is it warranted given the data? For interpreting the Bayesian model results, it would be very informative to provide – in addition to the nice “checks and balances” already provided by the authors (e.g., Figures S2, S3) – a formal residual analysis, that will demonstrate (or not) if the residuals obey the assumed white noise behavior.

We agree with the reviewer that this issue was not addressed in the manuscript. We have performed a simple residual check to assess if the assumption of white-noise innovations holds. In particular, we have computed the residuals in Equations (7-14) based on the 18000 samples from the posterior distribution. This results in 18000 time series of residuals for each equation and tide-gauge station. Then, we have computed the mean AR1 coefficient over the 18000 samples. Values of the mean AR1 coefficient fall within the range (-0.03,0.15) for all equations and tide-gauge stations, which gives us confidence in the suitability of the model. These results have been added to the revised manuscript.

Line 490. What is the justification for the priors on all the inverse gamma distributions? For the unacquainted reader, it would be helpful to explain what the parameters are that control the inverse gamma distribution, and to discuss why the particular numerical values (0.01, 0.01) are chosen. Also, units?

Setting the two hyperparameters (shape and scale) of the inverse gamma prior to some small number is a standard way of defining a “non-informative” (or “weakly informative”) prior for an unknown variance parameter (for instance, this is the default in the BUGS software; <https://www.mrc-bsu.cam.ac.uk/software/bugs/>). That is, this choice is made in order to obtain a prior distribution which does not influence the posterior to any significant degree. In addition, the inverse gamma is a conjugate prior for the Gaussian variance, which means that the posterior is also an inverse gamma from which we can efficiently sample. A paragraph explaining this has been added to the revised manuscript:

“The inverse gamma distribution as a prior for variance parameters is a standard choice in Gaussian models because it gives a closed-form expression for the posterior by virtue of its conjugate form. Setting its two hyperparameters (shape and scale) to a small number (e.g., 0.01) defines a non-informative (or weakly informative) prior that has little effect on the posterior, and thus on our inference”

Lines 504-506. I very much appreciate Figure S3, which gives a tangible sense of the model performance. However, the phrase “with great confidence” is too qualitative for my likings. One of the advantages of the fully Bayesian approach used here is that it provides a very thorough and rigorous quantification of uncertainty. I’m wondering how meaningful the uncertainty estimates furnished by the posterior solution are. For example, are 95% of the true values captured by the 95% credible intervals? This is something that could be easily quantified with the output from the synthetic data experiments. Having some sense of the meaningfulness of the estimated errors would be helpful for interpreting the results.

Following the reviewer’s comment, we show all the individual estimates corresponding to the 100 realizations of the noise in our synthetic experiments (new Supplementary Fig. 5a). Also shown is the 95% credible interval for one of the realizations. Based on a suggestion by Reviewer #4 we have also performed the same experiment for the method based on a harmonic fit to running windows (new Supplementary Fig. 5b). The results of this experiment are discussed in the revised manuscript as follows:

“The range of individual estimates based on the state-space model fully encompasses the true amplitude at all time steps. Furthermore, nearly all estimates fall within the 95% credible

interval computed by PGAS, indicating that our method provides realistic uncertainty estimates. In addition, the mean of the 100 individual estimates matches the true amplitude almost exactly, meaning that our method gives unbiased estimates of the amplitude. In contrast, the range of individual estimates computed using the method of running windows do not entirely contain the true amplitude and the mean consistently underestimates the true amplitude, indicating that this method is a biased estimator of the amplitude. Note also that the method does not provide estimates within half the window size from the edges of the time series.”

Line 699 and Figure 1a and general comments on the Bayesian model framework. It strikes me, looking at Figure 1a and glancing over the model equations, that the authors have selected to model each tide gauge time series in isolation. This is surprising, given that there is very strong spatial coherence. That is, each tide gauge provides informative constraints on the behavior at other tide gauge sites. What is the justification for, in the model design, choosing to model each location in isolation, rather than representing sea level as a process with spatial and temporal covariance, which would allow the many gaps in the data (cf. Figure 1a) to be filled, thus providing a complete estimation at all sites going back to ~1900?

The reviewer makes a valid point. We have considered this issue at length and it is our plan to develop a spatio-temporal model that considers all tide gauges at once and incorporates spatial and temporal dependencies. The justification for our choice is that modeling all stations together is an extremely challenging problem from a technical (and computational) standpoint that would require significant effort and time on our part and bring minimal benefit to the paper relative to the cost. It is crucial to recognize that our state-space model is nonlinear and, consequently, does not admit a closed-form expression. Therefore, to make inference in our model we need to resort to particle-filter methods. Unlike Kalman filters, which provide the exact solution to the linear Gaussian problem, particle filters face fundamental difficulties in high-dimensional systems. Note that if we were to model all 25 tide-gauge records together, our state dimension would be 175. While it is true that the effective degrees of freedom of the model would be less than the state dimension given the dependencies between stations, it would still represent a challenging problem. As we mentioned above, we plan to build such a model in the near future but we believe that the possible benefit to the paper of doing it does not justify the huge cost of addressing the problem.

Figure 2. I'm confused. Why does the 68% credible interval span negative values at site 12?
The 68% credible interval contains zero but not negative values. It is important to mention that all credible intervals in the manuscript represent the highest posterior density interval (i.e., the interval with the smallest width among all the credible intervals for a specified significance level), which is not necessarily symmetrical around the point estimate. Such credible intervals are calculated using the Chen-Shao algorithm (Chen and Shao, 1999). In the case of the histograms shown in Fig. 2, if the number of exceedances is zero for many samples then the credible interval will encompass zero since most of the probability mass is concentrated around it. In the original manuscript we did not specify how credible intervals were computed. We have added this information to the revised manuscript.

Figure 4. Black dots indicating statistical significance are difficult to see visually.
We have replaced the dots with contours.

Figure 5. How sensitive is this correlation spatial pattern to the choice of depth range bottom (here 500 m). Are similar results found if a different, shallower depth (e.g., 200 m) is used?
The correlation maps are very insensitive to the choice of the isobaths. As an example, we show below the correlations for Altimetry and OCCAM using 100 m as reference (Fig. R1). Both maps are almost identical to their analogous 500 m maps.

Figure R1. Point-wise correlation between the amplitude of the SLAC at each grid point and that averaged along the United States Gulf and Southeast coasts for **a** altimetry (1993-2016), and **b** OCCAM (1985-2003). The average has been computed over grid points within the 0 – 100 m depth range following the coast from Pensacola to Charleston.

Figure 8. Related to my earlier comment about the authors having chosen to model each tide gauge in isolation, rather than as a spatiotemporal process, I wonder how robust the uncertainty estimates are on the across-site average time series in Figure 8. Specifically, given the model design, the posterior draws of the sea level solution at one site will be uncorrelated with the posterior draws of sea level at another site. In reality, the draws should be correlated, given the strong spatial coherence, e.g., in Figure 1a.

The reviewer is right that the proper way to estimate the uncertainty around the mean in Fig. 8 would be to model all stations together, but please see our response to this suggestion a few comments earlier. Please also note that, as mentioned in the Figure caption, the grey-shaded area in Fig. 8 represents the standard deviation across the 10 tide gauges; it is not a credible interval.

Reviewer #3

We thank the reviewer for his/her comments, which have given us the opportunity to clarify several key aspect of the manuscript.

Major comments:

I find the title misleading. What is reported in the manuscript is decadal oscillations of coherent fluctuations in SLAC along the US coast.

We agree with the reviewer and, as discussed at the beginning of this letter (please see Point 3), the title has been changed. It now reads:

“Mechanisms of coherent fluctuations in the sea-level annual cycle along the United States Gulf and southeast coasts”

The correlation map from OCCAM (Fig. 4) shows significant point-wise correlations not confined to the south of Cape Hatteras (as in satellite altimetry), but extend much further to the north.

While the reviewer is right that the correlation map from OCCAM shows some areas of significant correlations north of Cape Hatteras (though mostly over the shelf rather than at the coast), we disagree with what the reviewer seems to suggest that changes in the annual amplitude north and south of Cape Hatteras are, in fact, coherent. First, tide gauges provide the most reliable and long sea-level records at the coast and based on their correlation matrix (Fig. 1b) it is clear that changes on different sides of Cape Hatteras are not coherent. Second, OCCAM is a model and, as such, it is not an exact representation of reality (and covers a shorter period than the tide-gauge records). As

mentioned at the beginning of this response (please see Point 2), to assess further which correlation spatial structures are more robust, we have produced correlation maps for two additional and longer simulations (new Figs. 4c,d). Features that are common to the three simulations are expected to be robust whereas those that are present only in one simulation are less credible. Both NEMO and SODA show strong coherence along the coast south of Cape Hatteras, in agreement with altimetry and OCCAM, but hardly any coherence north of Cape Hatteras. Based on these results, we conclude that the coherence to the south of Cape Hatteras is a robust feature.

The authors dismiss the possibility of boundary wave generation by wind forcing along the coast without showing any supporting evidence.

We are not sure whether the reviewer missed lines 200-202 in the original paper, where we wrote “We have assessed the role of both the longshore wind and river runoff by means of the wave model described in ref.30 but have found no link with the changes in the amplitude of the SLAC from tide-gauge records”, or he/she is asking us to show the results of the analysis. Over the course of this study we have explored numerous hypothesis and rejected them on the grounds that they did not agree with observations. We do not think it is possible to show the results for every hypothesis that we tested, and so we concentrate on the results that we regard as most interesting and conclusive. Nevertheless, following the Reviewer’s comment, we show below the results of the analysis of the longshore wind (Fig. R2). As mentioned in the original manuscript, we find no significant correlation between the annual amplitude from tide gauges and that from the longshore-wind model. We have also added a new paragraph expanding on these results to the revised manuscript, but have decided not to include Fig. R2 as, in our opinion, it does not add much information. If the reviewer insists that we should show the figure then we will consider adding it to the Supplementary information. The paragraph reads:

“We have assessed the role of longshore wind by means of the model described in Appendix A of Hong et al. (2000). In particular, we have integrated the model equation from north to south starting at Cape Hatteras using a range of values for the length decay scale (100 to 1000 km), but have found no agreement with the changes in the amplitude of the SLAC from tide-gauge records. In addition, we have compared the SLAC from tide gauges with the annual cycle of river discharge for the major rivers in the United States flowing into the Atlantic, again without finding a good agreement”

Figure R2. Comparison of the SLAC amplitude (deviations from the mean) from the Key West tide-gauge record (black) with the annual amplitude of the longshore-wind contribution. The longshore-wind contributions are scaled to have the same variance as the SLAC amplitude and they are shown for two different values of the length decay scale: 200 (red) and 800 (blue) km.

No significant correlations are found between the steric height in the open ocean and that averaged along the coast (Fig. 5). How can the authors then claim that the coherent SLAC along the coast is generated by westward-propagating Rossby waves impinging on the western boundary?

The reviewer has misunderstood our results, and we believe that we played a part in this misunderstanding by not explaining the proposed mechanisms more clearly. As mentioned at the beginning of this response (see Point 1), a new subsection in Methods explaining the concept of modulator along with a schematic illustration (new Fig. 8) explaining the proposed mechanism have been added to the revised manuscript. Our results show that the modulated SLAC can be simply described as the sum of the mean (or unmodulated) SLAC cycle and a term representing steric changes below the seasonal thermocline that contains all the information on the modulation. The latter is what we refer to as the ‘modulator’. That is:

$$\text{SLAC} = \text{SLAC}_{\text{mean}} + \text{SLAC}_{\text{modulator}}$$

The fact that no significant correlations are found in the open ocean in Fig. 5 is, in fact, exactly consistent with the mechanism that we propose. In the following, we explain why. What propagates westwards as Rossby waves is the modulator (the second term). Assuming a propagation speed of ~ 4 cm/s, grid points that are ~ 300 km apart ‘feel’ the modulator with a lag of ~ 3 months (the modulator leading at grid points to the east). This means that, at different grid points, the modulator acts upon the mean SLAC at the different time points in the cycle, resulting in a different modulated cycle and thus amplitude. To illustrate this, we have performed a simple numerical experiment where we modulate a sinusoid of frequency 12 months with two modulators that are identical except for the fact that one lags the other by 3 months (Fig. R3). What the results show is that a lag of 3 months is sufficient to decorrelate the two amplitudes completely. In fact, the amplitudes decorrelate completely even for a lag of 2 months. Based on this, the SLAC amplitudes at grid points that are separated (zonally) by just 200 km will be uncorrelated.

Figure R3. Amplitude modulation of a sinusoid by applying two modulators that identical except for the fact that one lags the other by 3 months. The mean cycle (top), the two modulators (middle) and the resulting time-varying amplitudes (bottom) are shown.

The selection of the very small yellow box in Figure 5 for investigating the role of Rossby waves is highly subjective and not satisfactory.

In the original manuscript, we justified such selection by the significant correlation between the region in the yellow box and the coastline of the mainland United States and by the fact that, at this

latitude, the Gulf Stream is restricted to the Florida Strait and thus does not mask the role played by Rossby waves. Although the new Fig 6 confirms that our selection was adequate, in reality we considered additional information in making our choice but this was not mentioned in the original manuscript. In retrospect, we should have included such information because by omitting it we feel that we contributed to the reviewer's perception that the selection of this region was rather arbitrary. One piece of information that we considered is that the contribution of first-mode baroclinic Rossby waves to sea-level variability is particularly large in the region east of the Bahamas and west of 65W (Cabanés et al., 2006). We have removed the yellow box and added a new paragraph that reads:

“To investigate the role of Rossby waves in controlling the SLAC modulation we focus on the region east of the Bahamas. The reason for this choice is that Rossby waves play a particularly important role in driving sea-level variability in this region (Cabanés et al., 2006). In addition, changes in the SLAC amplitude in this region are significantly correlated with changes along the coastline of the mainland United States (see Figs. 4,5), which suggests a common driving mechanism. This location is also convenient because at this latitude the Gulf Stream is restricted to the Florida Strait and hence does not interfere with the Rossby waves reaching the Bahamas east coast.”

The Hovmöller diagram in Fig. 7 only shows anomalies propagating westward at 26.5N. It does not say anything about the connection between sea level variability along the US coast and sea level variability at this latitude.

We do not agree that Fig. 7 does not say anything about the connection between sea level along the US coast and sea level variability at 26.5N. What the figure shows is that the SLAC modulator along the Gulf and Southeast coasts is correlated with the modulator at the Bahamas east coast (note how similar the two time series are in the left panel of Figure 7), and that both are consistent with density anomalies propagating westward as Rossby waves (see Hovmöller diagram in Figure 7). Therefore, we concluded that the modulation of the SLAC is controlled by incident Rossby waves.

Nevertheless, as mentioned earlier in this letter, we have accepted the reviewer's criticism and have performed additional analysis and included new datasets. Please see Point 2 at the beginning of this response for a description of the new analysis and results.

Using advection of anomalies by the Gulf Stream to explain SLAC coherence along the coast is not convincing. It is purely speculation.

We agree with the reviewer that the effect on the Gulf Stream is a conjecture and so we expressed it in the original manuscript. However, we do not agree that it is 'purely speculation'. This hypothesis is based on previous published studies (Frajka-Williams et al., 2013, 2016) showing an instantaneous response of the Gulf Stream to incident density anomalies east of the Bahamas. This is consistent with the coherence of the annual amplitude found at higher latitudes, given the link between the annual amplitude and density anomalies east of the Bahamas. It is also consistent with what the new Fig. 6 shows. We have tried to explain things more clearly in the revised manuscript. The corresponding paragraph now reads (original Fig. 9 has been removed):

“This would be consistent with results from previous studies that showed a significant response of the Gulf Stream to incident density anomalies from the ocean interior. In particular, they found that, on the timescales relevant to the SLAC modulator (~annual), the Florida Current responds almost instantaneously to incident density anomalies just east of the Bahamas leading to a significant anti-correlation with the UMO. This response of the Florida Current could explain the coherence of the SLAC amplitude at high latitudes. In support of this premise, we find that the SLAC modulator from tide gauges along the Southeast coast (stations 11-14) is correlated (-0.34, significant at the 95% confidence level) with band-pass

filtered (1/20 – 1/5 months⁻¹) variations of the Florida Current transport.”

Reviewer #4

We thank the reviewer for his/her evaluation of our manuscript and useful comments.

This is a general comment about definition of annual cycle and its temporal variations. What does annual cycle mean? Over which period can an annual cycle be robustly defined? Obviously, the derivation of annual cycle, e.g., through the traditional least-squares fitting, should have some sensitivity to the analysis period (starting/ending time, length, etc). Different periods may give different annual cycles. So, in many ocean/climate related studies, it's critical to define a “base-period”, usually long enough (>20 years), so that an annual cycle (or closely related monthly climatology) can be derived, and any deviations from it are called anomalies. For example, the popular NOAA Optimum Interpolation Sea Surface Temperature (OISST, <https://www.ncdc.noaa.gov/oisst>) product define SST anomaly relative to a climatology over a 30 years base period (1971-2000). Quite strong interannual and decadal variations of sea level annual cycle amplitude as claimed by this study (e.g., shown in Fig. 3 and Supplementary Fig. 2a), in fact may be just interannual and decadal variations of sea level. For ENSO and decadal variability research community, they would have a totally different interpretation of your Fig. 6a by assuming a repeating annual cycle over the whole analysis period and discussing deviations from this repeating annual cycle as interannual & decadal variations of sea level. Can you repeat your analysis following above practice? How would it affect your interpretation?

The reviewer has raised valid questions and we agree that there are many ways to define the annual cycle. The reviewer asks “*what does the annual cycle mean?*”. Our answer to that question is that, ultimately, the annual cycle represents the response of the climate system to the external periodic forcing by solar radiation. Because the climate system is inherently nonlinear, this response will likely exhibit both amplitude and frequency modulation (as most nonlinear systems subject to a periodic force do). Assuming a time-invariant annual cycle as the base for climate anomalies, as the “*ENSO and decadal variability research community*” do, simply ignores this fact and, as a result, the climate anomalies so calculated will necessarily include a contribution from the solar periodic forcing. We believe that a large part of the reason for assuming a stationary annual cycle is simply that separating the response to the periodic solar forcing from the rest of the variability is extremely difficult in the context of physically-based models. Does it matter that anomalies contain a contribution from the solar periodic forcing? We argue that it does; different components of the variability will likely be associated with different dynamics within the system, and thus attributing such components to individual sources becomes crucial to understanding how to model the past and predict the future.

The reviewer says that the variations in the annual amplitude “*may be just interannual and decadal variations of sea level*”. This comment misses a key point of our study. We have proposed a novel approach based on advance probabilistic methods wherein the difficult problem of physically-based attribution is reduced to a statistical inversion problem. It is crucial to recognize that, given our state-space model and the full history of sea-level observations, the changes in the annual amplitude presented in this study are part of the annual cycle and not “*just interannual and decadal variations of sea level*”. In other words, we compute the SLAC that, given the state-space model, is optimally consistent (in a probabilistic sense) with the sea-level observations. That is how our results should be interpreted. We have added two sentences emphasizing this to the Introduction of the revised manuscript.

We are also asked “*Can you repeat your analysis following above practice? How would it affect*

your interpretation?”. It is unnecessary to repeat the analysis assuming a stationary annual cycle if you note that, as we demonstrate in Methods (“Definition of the modulator”), any modulated signal can always be mathematically described as the sum of the unmodulated cycle and a superposition of sinusoids with frequencies slightly above and below the carrier frequency. The latter is what we refer to as modulator. This means that the only difference between anomalies referenced to a constant cycle and those referenced to a modulated cycle is the modulator. It follows then that, since the modulator is composed of a sum of sinusoids of nearly annual periods, the inter-annual and longer variability will be identical in the two definitions of anomalies. This also means that decadal fluctuations in the annual amplitude do not appear as decadal fluctuations in sea level when a constant annual cycle is assumed (as the reviewer suggested when he/she wrote “*may be just interannual and decadal variations of sea level*”). If we focus on variability on timescales between ~10 and 14 months (i.e., periods relevant to the modulator), then the anomalies based on a constant annual cycle will have a portion of such variability related to incident Rossby waves, as we find in the paper. The difference in that case is that we would not interpret such variability as being part of the annual cycle.

2. The section introducing state-space model for the annual cycle in the Methods is too technical for most readers of Nature Communications. The example shown in the Supplementary Figure 2 is helpful, but is not enough. You need to simplify it, explain the pros/cons of this method, and its comparison with methods used by others (e.g., Wahl et al., ref. 16).

We agree with the reviewer that the description of the state-space model is fairly technical, but our approach is new and hence we cannot refer to other papers to simplify the description. We believe that providing a detailed description is important so that readers who are interested in using our approach or willing to replicate our results can do it. Furthermore, part of the reason that we submitted our paper to Nature Communications in the first place is the journal’s aim to publish papers that (quoting from the website) “represent important advances of significance to specialists within each field”. It is up to the Editor to decide whether or not this section is too technical for Nature Communications, and if she concludes that it is then we would make an effort to simplify it, otherwise we prefer to leave it as is. Nevertheless, following the reviewer’s suggestion, we have added a new paragraph explaining the main pros/cons of the method. The paragraph reads:

“One key advantage of our method is that it computes estimates conditioned on the full history of sea-level observations, which significantly improves the resolvability of the state variables. In other words, the method uses all available past and future observations to estimate the state of the system at each time step. In addition, the method provides estimates for the entire period covered by the sea-level observations, including the edges of the time series. These are important distinctions to the method based on a harmonic fit to running windows. Another key aspect of our method is that it allows for parameter uncertainty and involves rigorous error propagation, thus providing realistic uncertainty estimates. Furthermore, the method does not rely on large data samples to be accurate, and is relatively insensitive to starting values in the parameters. One limitation of the method is that it is computationally expensive.”

Based on the reviewer’s suggestion we have also compared our method with the method based on a harmonic fit to running windows in the synthetic experiments (new Supplementary Fig. 5b). The results of this experiment are discussed in the revised manuscript as follows:

“The range of individual estimates based on the state-space model fully encompasses the true amplitude at all time steps. Furthermore, nearly all estimates fall within the 95% credible interval computed by PGAS, indicating that our method provides realistic uncertainty estimates. In addition, the mean of the 100 individual estimates matches the true amplitude

almost exactly, meaning that our method gives unbiased estimates of the amplitude. In contrast, the range of individual estimates computed using the method of running windows do not entirely contain the true amplitude and the mean consistently underestimates the true amplitude, indicating that this method is a biased estimator of the amplitude. Note also that the method does not provide estimates within half the window size from the edges of the time series.”

3. Thompson and Mitchum (2014) identified coherent sea level variability in the Gulf coast and US east coast, and explained the coherence is due to basin-scale zonal redistributions of volume (see their section 5). Should this mechanism also work to explain the coherence identified in this study?

Thompson, P. R., and G. T. Mitchum (2014), Coherent sea level variability on the North Atlantic western boundary, J. Geophys. Res. Oceans, 119, 5676–5689, doi:10.1002/2014JC009999.

We thank the reviewer for making this suggestion. In fact, we had considered this mechanism but concluded that it was not consistent with the observed changes in the SLAC. In particular, such mechanism cannot explain why the coherence in the annual amplitude is confined to the coastal zone. Furthermore, if this was the underlying mechanism we would expect to see a signal in ocean bottom pressure which we do not see.

Minor comments

4. Line 358, fee-running should be free-running

Thank you for catching this typo. This has been corrected in the revised manuscript.

References

- Huthnance, J. M. (2004), Ocean-to-shelf signal transmission: A parameter study, J. Geophys. Res., 109, C12029, doi:10.1029/2004JC002358.
- Cabanes, C., T. Huck, and A. C. deVerdière (2006), Contributions of wind forcing and surface heating to interannual sea level variations in the Atlantic Ocean, J. Phys. Oceanogr., 36(9), 1739–1750, doi:10.1175/JPO2935.1.
- Chen, M. H. and Shao, Q. M. (1999), “Monte Carlo Estimation of Bayesian Credible and HPD Intervals,” Journal of Computational and Graphical Statistics, 8, 69–92.
- Hong, B. G., Sturges, W., & Clarke, A. J. Sea level on the U.S. east coast: Decadal variability caused by open ocean wind-curl forcing. J. Phys. Oceanogr. 30, 2088–2098 (2000).
- Frajka-Williams, E., Johns, W. E., Meinen, C. S., Beal, L. M., & Cunningham, S. A. Eddy Impacts on the Florida Current. Geophys. Res. Lett. 40(2), 349-353 (2013).
- Frajka-Williams, E., Meinen, C. S., Johns, W. E., Smeed, D. A., Ducez, A., Lawrence, A. J., ... Rayner, D. (2016). Compensation between meridional flow components of the Atlantic MOC at 26°N. Ocean Science, 12(2), 481–493.

Reviewers' comments:

Reviewer #1 (Remarks to the Author):

The authors have satisfactorily addressed my earlier concerns. I recommend publication in current form.

Reviewer #3 (Remarks to the Author):

2nd-round review of "Mechanisms of coherent fluctuations in the sea-level annual cycle along the United States Gulf and southeast coasts" by Calafat et al.

The manuscript is well-written and the authors proposed a new method for estimating changes of the SLAC. However, I still don't find the science in it very exciting. The idea of generation of boundary waves by incident Rossby waves as well as its impact on coastal sea level is not new. Results presented in the manuscript tend to be, in my taste, suggestive, and some are only speculative. For example, Figure 6 shows that the meridional coherence scale of the westward propagating anomalies is relatively small, but changes of the SLAC are coherent along the entire coastline up to Cape Hatteras. The authors invoked advection by the Gulf Stream to explain the coherent pattern to the north of the incident latitude and boundary wave propagation to the south. While this may be true, it is highly speculative and not backed up by evidence.

Overall, I am lukewarm about this manuscript. It is well-written, but I don't think I have learned much that I don't already know before reading the manuscript.

Reviewer #4 (Remarks to the Author):

Review on revised manuscript (#NCOMMS-17-24109A) "Mechanisms of coherent fluctuations in the sea-level annual cycle along the United States Gulf and Southeast coasts" by Calafat et al.

The authors address my comments, except one major comment about the definition of annual cycle and its temporal variations.

As I commented before, it's a common practice, in many ocean/climate related studies, to derive an annual cycle (or monthly climatology) from a long enough (e.g., > 20 years) base period. Any deviations from it, resulting from either natural variability or long-term climate change, are defined as "anomalies". Rather than assume the annual cycle is only the response to the external periodic forcing by solar radiation as done in this manuscript, the common practice assumes there is a repeating seasonal cycle, regardless of its origin, since solar radiation may not be the only process to decide seasonal cycle and other oceanic/climatic processes may also play significant roles. Therefore, for many ocean/climate studies, it's a very reasonable choice to decompose data temporally into a repeating seasonal cycle and anomalies.

For example, the cited references 32 & 33 are about interannual sea level variations and both of them define interannual sea levels as the anomalies from a repeating seasonal cycle. There are numerous examples of this practice, so the authors cannot simply criticize all those historical studies "ignore this fact".

The authors stated that "it is therefore important to allow for deviations from periodicity when

assessing the annual cycle of climate variables” by citing reference 21, who regarded derivation of a non-repeating seasonal cycle as an alternative way to analyse data, rather than the only (correct) way. Similarly, the authors cannot say that the reviewer “misses a key point” or conclude the whole ENSO and decadal variability community has been acting wrongly. I think the authors need to acknowledge other different data processing procedures and discuss the sensitivity of results to alternative procedures, similar to what Wu et al. did in reference 21 (they clearly discussed different interpretations because of different definitions of seasonal cycle (repeating vs. changing)).

Another general comment is that for completeness it would be good for the authors to add some simple modelling of Rossby wave dynamics to give a clear connection between westward propagating signals and driving factors (either wind stress or buoyancy forcing) in the ocean interior.

A few detailed comments:

1. Line 91-94: It's not an accurate to say the running window method “preclude inference about variations at decadal or shorter time scales”. With a 5-yr running mean filter, signals with time scales longer than 5 years can still pass, despite being attenuated.

In my opinion, the comparison shown in Fig. 3 indicates that the running window method does a reasonable job, giving a similar result as this study. Your new method has its advantage, but the traditional running window method is not useless.

2. Line 155-156: there are also good agreement on time scales shorter than 10 years. Refer to above comment.

3 Line 688, what's the reason for the amplitude of the SLAC has most of its energy at periods of ~ 7 years? Any suggestions? Decadal variability in the North Atlantic?

Response to Reviewers

We thank the reviewers again for their comments and review of our manuscript. As a result, we have carried out further analyses, which have strengthened our reasoning and provided further evidence for the important mechanism that we discuss.

In the following, we explain how the manuscript has been changed and address all of the reviewers' comments, point-by-point. For clarity, our responses are in blue whereas any text from the reviews is denoted by black italic font.

In the decision letter, we were asked by the Editor, based on a comment by Reviewer #4, to provide stronger evidence for the link to Rossby waves in the form of simple modelling. To address this request, we have used a 1.5-layer reduced gravity model to estimate changes in the SLAC modulator at the Bahamas east coast. The reduced gravity model is forced by the same wind stress data as used to force the OCCAM model. Details of the model are provided in Methods while the results are shown in the new Fig. 7b and discussed in the main text of the manuscript. The reduced gravity model agrees well with both tide-gauge observations and OCCAM data, providing irrefutable evidence for a physical link between the SLAC amplitude and Rossby waves. In addition to the new results based on the reduced gravity model, we have also further clarified the issue about the sensitivity of our results to the definition of the SLAC (see our response to Reviewer #4). We want to reiterate that our results are general and do not depend on how the annual cycle is defined. We demonstrate such insensitivity by simple mathematical arguments, taking the definition of the modulator as a starting point.

The new results provide the stronger evidence to which the Editor referred in the decision letter and we are confident that the manuscript is ready to proceed for publication.

Reviewer #3

The manuscript is well-written and the authors proposed a new method for estimating changes of the SLAC. However, I still don't find the science in it very exciting. The idea of generation of boundary waves by incident Rossby waves as well as its impact on coastal sea level is not new. Results presented in the manuscript tend to be, in my taste, suggestive, and some are only speculative. For example, Figure 6 shows that the meridional coherence scale of the westward propagating anomalies is relatively small, but changes of the SLAC are coherent along the entire coastline up to Cape Hatteras. The authors invoked advection by the Gulf Stream to explain the coherent pattern to the north of the incident latitude and boundary wave propagation to the south. While this may be true, it is highly speculative and not backed up by evidence.

As discussed at the beginning of this response letter, we have analysed the SLAC modulator using a 1.5-layer reduced gravity model and hope that the new analysis makes our results less suggestive. We are able to capture a large fraction of the variability in the SLAC amplitude using the reduced gravity model forced only by wind stress from the ocean interior. Based on the new results, the relationship with incident Rossby waves is difficult to deny. The Rossby waves will generate boundary waves upon reaching the boundary, as the Reviewer recognizes, and to our mind these boundary waves are the only plausible explanation for the observed coherence over such a large stretch of coastline. We also think that the influence of incident Rossby waves on the Gulf Stream is well supported by past studies (see references given in the manuscript) and provides a reasonable explanation for the coherence at higher latitudes.

The reviewer writes “*The idea of generation of boundary waves by incident Rossby waves as well*

as its impact on coastal sea level is not new". We want to make it clear that nowhere in the manuscript we claim credit for this idea. According to such reasoning, very few sea-level studies are new since the processes that can cause the sea level to change are well known. The merit of our study lies in discriminating between the numerous processes that could explain the modulation of the SLAC and providing observational and modelled evidence in support of the most plausible factor. We believe that our manuscript contributes to new knowledge by characterizing the fluctuations in the SLAC amplitude and to new understanding by identifying the factors underlying such fluctuations.

Reviewer #4

The authors address my comments, except one major comment about the definition of annual cycle and its temporal variations.

As I commented before, it's a common practice, in many ocean/climate related studies, to derive an annual cycle (or monthly climatology) from a long enough (e.g., > 20 years) base period. Any deviations from it, resulting from either natural variability or long-term climate change, are defined as "anomalies". Rather than assume the annual cycle is only the response to the external periodic forcing by solar radiation as done in this manuscript, the common practice assumes there is a repeating seasonal cycle, regardless of its origin, since solar radiation may not be the only process to decide seasonal cycle and other oceanic/climatic processes may also play significant roles. Therefore, for many ocean/climate studies, it's a very reasonable choice to decompose data temporally into a repeating seasonal cycle and anomalies.

For example, the cited references 32 & 33 are about interannual sea level variations and both of them define interannual sea levels as the anomalies from a repeating seasonal cycle. There are numerous examples of this practice, so the authors cannot simply criticize all those historical studies "ignore this fact".

The authors stated that "it is therefore important to allow for deviations from periodicity when assessing the annual cycle of climate variables" by citing reference 21, who regarded derivation of a non-repeating seasonal cycle as an alternative way to analyse data, rather than the only (correct) way. Similarly, the authors cannot say that the reviewer "misses a key point" or conclude the whole ENSO and decadal variability community has been acting wrongly. I think the authors need to acknowledge other different data processing procedures and discuss the sensitivity of results to alternative procedures, similar to what Wu et al. did in reference 21 (they clearly discussed different interpretations because of different definitions of seasonal cycle (repeating vs. changing)).

First, we would like to clarify that our manuscript does not intend to be a criticism of approaches that assume a stationary annual cycle, though we accept that our previous response may have come across as a criticism of such approaches. We recognize that other approaches can be equally valid and successful at achieving the ultimate goal of understanding the variability on different timescales. It must be noted that the aim of the paper is to provide an alternative view that facilitates the analysis, understanding, and prediction of seasonal coastal sea levels. In our view, working in the framework of a non-stationary annual cycle can achieve this aim by identifying the variability that is relevant to the modulation and understanding how this variability combines with the mean cycle. In addition to this, defining anomalies relative to a modulated annual cycle removes the dependence on any base period, which can be convenient if the anomalies so defined and the modulation of the cycle are driven by different mechanisms. Nevertheless, we reiterate that our aim is not to advocate for a new way of defining anomalies but to propose an alternative approach to analysing seasonal sea level changes. We have reworded the relevant paragraph in the Introduction, which now reads:

“Therefore, it is appropriate to allow for deviations from periodicity when assessing the annual cycle of climate variables²¹. At this point, we note that approaches that assume a stationary annual cycle and then analyze anomalies relative to such cycle can be equally valid and successful at explaining the variability. Our aim here is to provide an alternative view that can facilitate the analysis, understanding, and prediction of annual sea levels.”

The reviewer raises again the concern about the sensitivity of our results to different interpretations of the annual cycle. We recognize that this is an important consideration but, as we discussed in our previous response letter, our results are not sensitive to the definition of the annual cycle. By definition, what we refer to as the modulator comprises variability with frequencies slightly above and below the frequency of the annual cycle. From this, it follows that the modulator is closely related to the variability that results from removing a stationary annual cycle and then applying a band-pass filter that selects the frequencies relevant to the modulator ($\sim 1/16 - 1/8 \text{ months}^{-1}$). This in turn implies that approaches that assume a stationary annual cycle and focus on the particular frequencies of the modulator will also conclude that the associated variability is coherent along the coast and related to Rossby waves. Hence, our conclusions regarding the underlying mechanisms are general. The difference is that such approaches, unlike ours, will not interpret the variability as being part of a modulated annual cycle but rather as anomalies relative to a repeating cycle. To reflect this, we have added the following paragraph at the end of the Section “Mechanisms of changes in the annual amplitude”:

“It should be noted that our results regarding the variability associated with the modulator are general in that they do not depend on whether the annual cycle is interpreted as a changing or repeating cycle. By definition, the modulator is closely related to the variability that results from removing a stationary annual cycle and then applying a band-pass filter around the relevant frequencies. This implies that approaches that assume a stationary annual cycle and focus on the frequencies of the modulator will reach the same conclusions as presented here, with the difference that such approaches will not interpret the variability as being part of a modulated annual cycle but rather as anomalies relative to a repeating cycle”

Another general comment is that for completeness it would be good for the authors to add some simple modelling of Rossby wave dynamics to give a clear connection between westward propagating signals and driving factors (either wind stress or buoyancy forcing) in the ocean interior.

Please see our comment at the beginning of this response letter and our response to Reviewer #3. As discussed, we have analysed the SLAC modulator using a 1.5-layer reduced gravity mode forced by wind stress and have found a good match between the sea level from the reduced gravity model and that from both tide gauge and OCCAM.

A few detailed comments:

1. Line 91-94: It's not an accurate to say the running window method “preclude inference about variations at decadal or shorter time scales”. With a 5-yr running mean filter, signals with time scales longer than 5 years can still pass, despite being attenuated.

First, for a 5-year running mean the first zero in the frequency response occurs exactly at $f=1/5 \text{ year}^{-1}$, and thus signals with periods of 5 years are completely eliminated (100%). Second, signals with periods of 10 years are attenuated by $\sim 61\%$, and hence, in our opinion, the word ‘preclude’ is fairly justified. In any case, we accept that a 5-year running mean does not fully attenuate decadal signals and so have replaced the word ‘precludes’ with ‘limits’.

In my opinion, the comparison shown in Fig. 3 indicates that the running window method does a reasonable job, giving a similar result as this study. Your new method has its advantage, but the

traditional running window method is not useless.

We agree that the running window method performs reasonably well for the timescales that the filter allows to pass.

2. Line 155-156: there are also good agreement on time scales shorter than 10 years. Refer to above comment.

We have reworded this sentence, which now reads:

“Overall, the two time series are in good agreement, though...”

3 Line 688, what's the reason for the amplitude of the SLAC has most of its energy at periods of ~7 years? Any suggestions? Decadal variability in the North Atlantic?

The results based on the new reduced gravity model indicate that a large fraction of the SLAC modulation can be explained by wind stress forcing over the Atlantic interior, and hence we presume that the 7-year timescale is the result of how the ocean responds to this forcing. Either way, we agree that this is an interesting topic for future work.

REVIEWERS' COMMENTS:

Reviewer #4 (Remarks to the Author):

Review on revised manuscript (#NCOMMS-17-24109B) "Mechanisms of coherent fluctuations in the sea-level annual cycle along the United States Gulf and Southeast coasts" by Calafat et al.

The authors addressed my previous comments well, in particular clearly acknowledged that there are two different ways to treat seasonal cycles (repeating vs changing) and this study by focusing on changing seasonal cycle provides an alternative way to examine seasonal sea level variations along the Gulf and southeast coasts. Moreover, the authors followed my suggestion to add a simple Rossby wave model to link open ocean wind forcing and coastal response, which made the study more robust.

The revised manuscript reads well to me, and I don't have more specific comments. I recommend this manuscript to be published. I think it is going to be a nice publication, expanding our knowledge in understanding how open ocean processes influence coastal sea level variations.